# MCCE: A FRAMEWORK FOR MULTI-LLM COLLABORATIVE CO-EVOLUTION

## ABSTRACT

Multi-objective discrete optimization problems, such as molecular design, pose significant challenges due to their vast and unstructured combinatorial spaces. Traditional evolutionary algorithms often get trapped in local optima, while expert knowledge can provide crucial guidance for accelerating convergence. Large language models (LLMs) offer powerful priors and reasoning ability, making them natural optimizers when expert knowledge matters. However, closed-source LLMs, though strong in exploration, cannot update their parameters and thus cannot internalize experience. Conversely, smaller open models can be continually fine-tuned but lack broad knowledge and reasoning strength. We introduce Multi-LLM Collaborative Co-evolution (MCCE), a hybrid framework that unites a frozen closed-source LLM with a lightweight trainable model. The system maintains a trajectory memory of past search processes; the small model is progressively refined via reinforcement learning, with the two models jointly supporting and complementing each other in global exploration. Unlike model distillation, this process enhances the capabilities of both models through mutual inspiration. Experiments on multi-objective drug design benchmarks show that MCCE achieves state-of-the-art Pareto front quality and consistently outperforms baselines. These results highlight a new paradigm for enabling continual evolution in hybrid LLM systems, combining knowledge-driven exploration with experience-driven learning. The code of MCCE is available on https://anonymous.4open.science/r/MCCE_Anonymous-1F92

## 1 INTRODUCTION

Discrete optimization and multi-objective optimization problems are pervasive in real-world applications, ranging from logistics and scheduling to scientific discovery and molecular design (Sun et al., 2025). These problems are notoriously difficult due to their vast, high-dimensional, and unstructured search spaces. Traditional evolutionary algorithms, while widely adopted, often suffer from two critical limitations: (i) they are prone to premature convergence, getting trapped in local optima, and (ii) they struggle to maintain both diversity and quality in the candidate population. These limitations highlight the need for more adaptive, intelligent optimization frameworks.

The rise of Large Language Models (LLMs) opens a promising direction. With their strong reasoning ability and broad prior knowledge, LLMs can act as powerful operators for generating and refining candidate solutions (Zhao et al., 2025). However, their application in iterative optimization remains constrained. First, a single LLM tends to converge to its own distribution, reducing solution diversity across generations (Li et al., 2025; Luo et al., 2025; Gao et al., 2025b). Second, although retrieval-augmented generation (RAG) enables the injection of external knowledge through contextual retrieval, it is inherently limited by the size of the context window and lacks the ability to update model parameters. As a result, such systems cannot genuinely accumulate knowledge or learn from past experiences. These challenges highlight that effective optimization requires not only problem-solving capacity but also mechanisms for internalizing feedback and continuously evolving.To this end, we argue that parameter training is indispensable. Unlike static prompting or RAG, parameter updates enable a model to accumulate experience in a much deeper and more persistent way. Yet, this poses a dilemma: closed-source LLMs excel in reasoning and general knowledge but cannot be fine-tuned, whereas small open-source models are trainable but lack the broad capabilities of larger models. Relying solely on either side leads to inherent inefficiency and bottlenecks.

This motivates our proposed Multi-LLM Collaborative Co-evolution (MCCE) framework, a system where a frozen, closed-source LLM and a lightweight, trainable local model co-evolve through iterative collaboration. In each generation, the two models alternate as evolutionary operators: the closed-source LLM drives global exploration, while the local model learns from accumulated experiences to perform more targeted searches. Crucially, we design a feedback loop where the local model is periodically refined using breakthrough search trajectories, ensuring that knowledge is continually internalized and reused. Unlike traditional distillation, our framework establishes mutual inspiration between models—large models provide global guidance, while small models adaptively extend the search frontier through learning. Recent work such as ExLLM (Ran et al., 2025) has demonstrated the promise of using LLMs as evolutionary operators for multi-objective molecular design, combining in-context learning with prompt engineering to achieve strong results. However, these approaches still rely on a single frozen LLM, which limits their ability to accumulate experience through parameter updates and often leads to reduced diversity and premature convergence. In contrast, our MCCE framework explicitly addresses this gap by coupling a powerful but fixed closed-source LLM with a lightweight trainable model. This collaborative co-evolution not only preserves the broad reasoning and exploration capacity of large models, but also equips the system with a mechanism for continual learning and adaptation. By enabling mutual inspiration between heterogeneous models, MCCE overcomes the limitations of purely LLM-driven pipelines and establishes a more sustainable path toward scalable optimization.

The main contributions of this paper are:

1. **A collaborative co-evolution framework (MCCE).** We integrate closed-source LLMs with lightweight, trainable local models, combining the exploration capacity of large models with the adaptability of smaller models. This hybrid design is broadly applicable to discrete, multi-objective optimization tasks beyond drug discovery.

2. **An experience-driven learning paradigm.** We leverage breakthrough evolutionary trajectories as valuable experience, guiding the local model to identify promising search directions. This cooperative mechanism allows the global and local models to co-evolve, reinforcing each other's strengths over time.

3. **Demonstrated practical efficacy and extensibility.** Our framework achieves state-of-the-art performance in multi-objective drug design, highlighting its potential for real-world impact. Moreover, the paradigm is extensible to a wider range of scientific and engineering domains where structured optimization is critical.

## 2 RELATED WORK

### 2.1 MULTI-MODEL COLLABORATION

Recent studies highlight the promise of collective intelligence in enhancing reasoning and problem-solving through multiple LLMs (JIANG et al., 2025). For example, Misaki et al. (2025) propose an adaptive branching Monte Carlo Tree Search (MCTS) framework where multiple models cooperate to balance exploration and exploitation. Extending this collaborative paradigm to scientific discovery, Su et al. (2025) employ a multi-agent system to mimic human teamwork for generating and refining novel research ideas. By leveraging diverse model perspectives in the search process, these approaches significantly improve efficiency and robustness compared to using a single LLM. Beyond inference scaling, other works explore multi-agent or ensemble strategies. Yang et al. (2025) demonstrate that integrating diverse reasoning pathways improves search-based reasoning, while Gao et al. (2025a) show the benefits of cross-model collaboration in structure-based drug design. Similarly, ensemble methods such as Huang et al. (2024) and Wang et al. (2023) propose novel ways to combine outputs or probability distributions across heterogeneous LLMs. While most aforementioned methods treat models as static entities, recent works have begun to incorporate training into the collaborative loop. For instance, Wu et al. (2025b) introduce reinforcement fine-tuning to transform models from passive responders into active collaborators. regarding heterogeneous collaboration, Lu et al. (2025) fine-tune small models to orchestrate fixed LLMs for cost-effective data labeling, and Xu et al. (2025) train small models to decompose queries to assist black-box LLMs in retrieval tasks. However, these approaches typically limit parameter updates to a single side of the collaboration or distinct functional modules. In contrast, our work emphasizes dynamic co-

evolution, where both large and small models jointly learn and evolve on the same optimization task through shared experience.

## 2.2 Experience Learning

The ability of LLM-based agents to continuously learn from experience has been recognized as a critical step toward AGI (Zheng et al., 2025). Several approaches explore reinforcement learning (RL) as a means of improving reasoning. For example, ML-agent (Liu et al., 2025b) apply online RL for autonomous machine learning engineering, while CALM (Huang et al., 2025) and Evo-Tune (Surina et al., 2025) combine RL with evolutionary search to refine heuristics and algorithms. However, traditional RL often struggles with the capability boundary of base models. Works such as RL-PLUS (Dong et al., 2025) and LUFFY (Yan et al., 2025) address this by introducing hybrid-policy optimization or off-policy guidance. Complementary strategies, including ReLIFT (Ma et al., 2025) and TAPO (Wu et al., 2025a), integrate supervised fine-tuning or structured external guidance to capture knowledge beyond the reach of RL. These methods show that a single LLM can incrementally improve through experience, but they remain limited by the inherent ceiling of one model. Our approach differs by enabling multi-model collaborative experience learning, where small models benefit from learning while also enriching the exploration capacity of larger models, forming a co-evolutionary loop.

## 2.3 Evolutionary Algorithms

A rapidly growing body of work explores integrating LLMs with evolutionary algorithms for optimization and design. For example, FunSearch (Romera-Paredes et al., 2024), EoH (Liu et al., 2024) and MEoH (Yao et al., 2025) demonstrate that LLMs can serve as generators for heuristics or algorithms in combinatorial optimization problems. Reflective mechanisms further enhance search efficiency, as seen in REEVO (Ye et al., 2024) and ML-master (Liu et al., 2025a), where memory or reflection guides iterative exploration. Evolutionary methods have also been applied in specialized domains, including prompt evolution for jailbreak attacks (Liu et al., 2023) or over-refusal mitigation (Wu et al., 2025c). More recent works such as Alphaevolve (Novikov et al., 2025) and Dat et al. (2025) introduce evaluator feedback loops, but still treat LLMs as static generators within the search process. Overall, while these studies validate the synergy between LLMs and evolutionary computation, they typically lack parameter-level adaptation or multi-model dynamics. Our contribution is to close this gap by combining evolutionary search with experience-driven training and collaborative co-evolution across models.

## 3 Preliminary

### 3.1 Reinforcement Learning (RL) and Direct Preference Optimization (DPO)

In Reinforcement Learning (RL), an agent learns a policy $\pi(a \mid s)$, which defines the probability of taking action $a$ given state $s$. The objective is to maximize the expected cumulative reward:

$$J(\pi) = \mathbb{E}_{\tau \sim \pi} \left[ \sum_{t=0}^{T} \gamma^t r(s_t, a_t) \right],$$ (1)

where $\tau = (s_0, a_0, \dots, s_T)$ is a trajectory, $r(s_t, a_t)$ is the reward at step $t$, and $\gamma \in (0, 1]$ is the discount factor.

Direct Preference Optimization (DPO) replaces explicit rewards with pairwise preferences over trajectories. Given a preferred trajectory $\tau^+$ and a dispreferred one $\tau^-$, the DPO loss is:

$$\mathcal{L}_{\text{DPO}}(\pi) = -\mathbb{E}_{(\tau^+, \tau^-)} \left[ \log \sigma \Big( \beta \Big( \log \frac{\pi(\tau^+)}{\pi_{\text{ref}}(\tau^+)} - \log \frac{\pi(\tau^-)}{\pi_{\text{ref}}(\tau^-)} \Big) \Big) \right],$$ (2)

where $\pi_{\text{ref}}$ is a frozen reference model, $\sigma$ is the sigmoid function, and $\beta$ controls preference sharpness.

## 3.2 SUPERVISED FINE-TUNING (SFT)

Supervised Fine-Tuning (SFT) adapts a pre-trained LLM by minimizing the negative log-likelihood (NLL) of reference outputs $y = (y_1, \ldots, y_T)$ given a prompt $x$:

$$\mathcal{L}_{\text{SFT}}(\theta) = -\sum_{t=1}^{T} \log p_\theta(y_t \mid x, y_{<t}). \tag{3}$$

This objective encourages the model to replicate high-quality, task-specific examples.

## 3.3 GENERATIVE FLOW NETWORKS (GFLOWNETS)

GFlowNets aim to generate diverse trajectories $\tau = (s_0 \rightarrow s_1 \rightarrow \cdots \rightarrow s_T)$ such that their probability is proportional to a reward function $R(s_T)$:

$$P_\theta(\tau) \propto R(s_T). \tag{4}$$

This is enforced through the flow matching constraint, ensuring that the incoming and outgoing flows at each state are balanced:

$$\sum_{s': \, s \rightarrow s'} F_\theta(s \rightarrow s') = \sum_{s'': \, s'' \rightarrow s} F_\theta(s'' \rightarrow s), \tag{5}$$

where $F_\theta(s \rightarrow s')$ is the probability flow along an edge.

# 4 METHODOLOGY

We propose Multi-LLM Collaborative Co-evolution (MCCE), a unified and general-purpose optimization framework for complex discrete problems, demonstrated here in molecular design. As shown in Figure 1, the system operates through an iterative collaboration between two distinct LLMs: a powerful but frozen model and a lightweight, trainable local model. The frozen LLM provides robust global exploration, while the local model continuously refines its policy by learning from successful search trajectories, forming a self-improving feedback loop. To validate MCCE, we adopt a challenging five-objective molecular optimization task, jointly targeting *QED*, *synthetic accessibility (SAscore)*, *DRD2 binding*, *GSK3β binding*, and *JNK3 binding*. This setting builds on recent benchmarks such as ExLLM (Ran et al., 2025) and MoLLEO (Wang et al., 2024), which emphasize that realistic drug discovery requires balancing multiple properties. While MoLLEO showed the benefit of LLM-based evolutionary operators, its evaluation was restricted to three objectives. By extending to five objectives, we align with prior work while pushing toward more realistic, high-dimensional challenges, providing a rigorous test of MCCE's adaptability.

## 4.1 OVERALL FRAMEWORK

The proposed MCCE framework operates in an iterative evolutionary loop, where large language models (LLMs) act as adaptive genetic operators. The overall process can be divided into four key stages: initialization, generation, evaluation, and update with learning.

**Stage 1: Initialization.** Let $\mathcal{P}_t$ denote the population pool at generation $t$, consisting of candidate molecules. The process begins with an initial population $\mathcal{P}_0$, which can be sampled either from an external database or generated by a pretrained LLM:

$$\mathcal{P}_0 = \{c_1, c_2, \ldots, c_M\}, \quad c_i \sim \pi_{\text{init}}(\cdot), \tag{6}$$

where $\pi_{\text{init}}$ represents the initialization distribution.

**Stage 2: Candidate Generation.** At each generation $t$, two parents $p_1, p_2 \in \mathcal{P}_t$ are selected according to a selection strategy (e.g., tournament or fitness-proportional selection). Given the pair $(p_1, p_2)$ and a task-specific prompt function $\text{prompt}(p_1, p_2)$, the LLM-based operator produces two new candidates:

$$(c_1, c_2) \sim \pi_{\text{LLM}}(\cdot \mid \text{prompt}(p_1, p_2)). \tag{7}$$

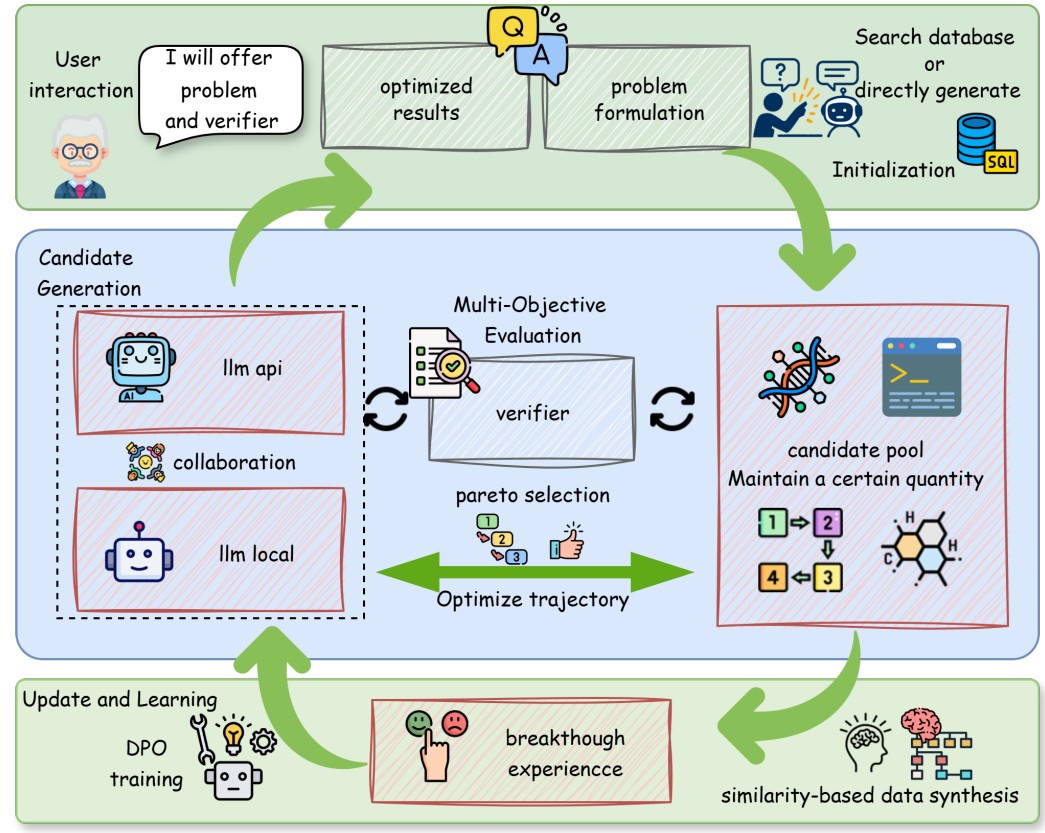

Figure 1: Overview of the proposed MCCE framework. The system begins with **user interaction** and **population initialization** based on the problem definition and evaluation criteria. In the **candidate generation** stage, a frozen API-based LLM and a trainable local LLM collaborate to propose new molecules. These are evaluated by the **multi-objective evaluation** module, which applies Pareto selection to maintain a balanced population, while breakthrough solutions are stored as experience. In the **update and learning** stage, similarity-based data synthesis constructs preference pairs from past trajectories, and the local model is refined via DPO training. This creates a self-improving feedback loop where global exploration (API LLM) and local adaptation (trainable LLM) co-evolve toward progressively optimized solutions.

Since each invocation of the operator generates exactly two candidates, constructing a full population of size $M$ requires

$$\frac{M}{2} \quad \text{generations of prompts.} \tag{8}$$

This process is repeated with different parent pairs until the entire offspring set is produced. The operator $\pi_{\text{LLM}}$ alternates between a frozen API model and a locally trainable model, thereby balancing *global exploration* (via frozen LLM) and *local adaptation* (via trainable LLM).

**Stage 3: Multi-Objective Evaluation.** Each generated candidate $c$ is evaluated using a multi-objective scoring function:

$$\mathbf{s}(c) = \big[s_1(c), s_2(c), \ldots, s_K(c)\big], \tag{9}$$

where $s_k(c)$ is the score under the $k$-th objective (e.g., drug-likeness, synthesizability, or binding affinity). All scores are normalized to a common scale:

$$\hat{s}_k(c) = \frac{s_k(c) - \mu_k}{\sigma_k}, \tag{10}$$

where $\mu_k$ and $\sigma_k$ are the mean and standard deviation of scores in the current population.

**Stage 4: Update and Learning.** The next-generation population $\mathcal{P}_{t+1}$ is formed by applying Pareto front selection, which preserves non-dominated solutions while maintaining diversity. Meanwhile, after every $N$ generated candidates, successful trajectories

$$(\text{prompt}(p_1, p_2) \ \rightarrow \ (c_1, c_2) \ \rightarrow \ \mathbf{s}(c_1), \mathbf{s}(c_2))$$

are stored as experience $\mathcal{D}$. This dataset is then used to refine the trainable LLM. Formally, the model parameters are updated as

$$\pi_{\text{LLM}} \ \longleftarrow \ \text{Update}(\pi_{\text{LLM}}, \mathcal{D}), \tag{11}$$

where $\text{Update}(\cdot)$ denotes an abstract learning procedure based on the accumulated experience. This establishes a closed-loop cycle of *generation–evaluation–learning–evolution*.

## 4.2 Which Training Paradigm Best Supports Experience-Driven Learning?

A central question in our framework is how to effectively refine the local model's policy through accumulated experience. To this end, we systematically explored several candidate training paradigms and evaluated their suitability for stabilizing learning while preserving the model's exploratory capacity. Our findings reveal critical limitations in conventional approaches:

**Supervised Fine-Tuning (SFT).** We first adopted SFT by treating "breakthrough" generations as positive training samples. Concretely, if a generated molecule achieved a score higher than all of its parents, the corresponding trajectory was labeled as effective data. However, this approach led to catastrophic forgetting: after training, the uniqueness of generated molecules dropped substantially. This indicates that the local model tended to memorize successful chemical formulas rather than internalize a generalizable exploration strategy, thereby losing its ability to propose genuinely novel solutions.

**Reinforcement Learning (RL).** Next, we experimented with reinforcement learning using the scoring function as the reward signal. In practice, this training proved highly unstable. Strong negative rewards for low-scoring molecules caused the model to collapse, as it struggled to infer the underlying reasons for the penalties and consequently lost its ability to generate valid candidates. The mapping between molecular structures and their scores is inherently unpredictable for an LLM, making explicit quantitative rewards unsuitable for stable RL training in this context.

**Direct Preference Optimization (DPO).** To overcome these issues, we adopted a DPO-based approach, which provides a more stable and sample-efficient training signal without requiring an explicit reward model. Initially, we constructed training pairs by contrasting high-scoring versus low-scoring molecules under the same prompt. However, we observed unstable loss oscillations: since identical prompts were associated with conflicting responses, the model often became confused. To address this, we developed a *similarity-based data synthesis* method, which ensures that preference pairs are constructed from structurally comparable molecules. This adjustment significantly improved both training stability and data efficiency. The details of this method are elaborated in Section 4.3.

## 4.3 Similarity-Based Data Synthesis

Our DPO training requires triplets of the form $(q, \tau^+, \tau^-)$ where $q$ is a query (prompt), $\tau^+$ is a preferred (chosen) trajectory and $\tau^-$ is a rejected trajectory. To construct such triplets stably and to mitigate distributional shift between the frozen API model and the local trainable model, we propose a similarity-based data synthesis pipeline. The pipeline proceeds in three phases: (1) collect candidate pool and compute similarity statistics, (2) filter and stratify candidates by score and similarity, (3) assemble DPO triplets with fallback rules.

**Notation.** Let $\mathcal{H} = \{q_1, q_2, \ldots, q_{|H|}\}$ be the historical prompts (ordered by time). For each prompt $q_j$ we have a set of generated candidates $\mathcal{C}_j = \{c_{j,1}, c_{j,2}, \ldots\}$, produced by either the frozen LLM or the local model during the recent evolution window. Let $s(c)$ denote the (multi-objective) score of candidate $c$ (we use a scalarized score or a ranking for stratification). Define a molecular similarity function $\text{sim}(c, q) \in [0, 1]$, computed by a fingerprint-based metric (e.g., Tanimoto on Morgan fingerprints) or any task-appropriate similarity $\phi(\cdot, \cdot)$.

**Phase 1 — similarity statistics.** Collect the similarity values across the considered history and models:

$$\mathcal{S} = \big\{\mathrm{sim}(c,q) \ : \ q \in \mathcal{H}, \ c \in \mathcal{C}_q \big\}.$$

Compute the empirical mean and standard deviation:

$$\mu \ = \ \frac{1}{|\mathcal{S}|} \sum_{x \in \mathcal{S}} x, \qquad \sigma \ = \ \sqrt{\frac{1}{|\mathcal{S}|} \sum_{x \in \mathcal{S}} (x - \mu)^2}. \tag{12}$$

We will use $(\mu, \sigma)$ as global similarity statistics to reduce distributional mismatch between models (both models' outputs contribute to $\mathcal{S}$).

Define a global similarity filter:

$$\mathcal{F} \ = \ \{ \, c \mid \mu - \sigma \leq \mathrm{sim}(c,q) \leq \mu + \sigma \, \}. \tag{13}$$

Only candidates in $\mathcal{F}$ are considered for DPO pair construction (this ensures candidates are within one standard deviation of the empirical similarity distribution).

**Phase 2 — score stratification and similarity windows.** Let $\alpha$ denote the quantile threshold used to form top/bottom pools (we use $\alpha = 0.3$ by default). Let $\mathcal{C}_{\mathrm{all}} = \bigcup_q \mathcal{C}_q$ and sort $\mathcal{C}_{\mathrm{all}}$ by score $s(\cdot)$. Define

$$\mathcal{T}_{\mathrm{high}} = \{ \, \text{top } \alpha \text{ fraction of } \mathcal{C}_{\mathrm{all}} \, \}, \qquad \mathcal{T}_{\mathrm{low}} = \{ \, \text{bottom } \alpha \text{ fraction of } \mathcal{C}_{\mathrm{all}} \, \}.$$

We further define nested similarity intervals (from strict to relaxed):

$$I_1 = [\mu + \tfrac{2}{3}\sigma, \ \mu + \sigma], \quad I_2 = [\mu + \tfrac{1}{3}\sigma, \ \mu + \sigma], \quad I_3 = [\mu, \ \mu + \sigma]. \tag{14}$$

These intervals prioritize chosen candidates that are both high-scoring and reasonably similar to the prompt (thus reducing contradictory prompt–response pairs that destabilize training).

**Phase 3 — per-prompt pair construction with fallback rules.** To construct stable DPO training triplets, we design a per-prompt pair construction algorithm that selects a preferred ($\tau^+$) and a rejected ($\tau^-$) candidate for each prompt $q$. As outlined in Algorithm 1, the procedure first filters candidates by similarity, then attempts to select $\tau^+$ from the high-score pool and $\tau^-$ from the low-score pool using progressively relaxed similarity intervals ($I_1 \rightarrow I_2 \rightarrow I_3$), and finally falls back to broader score ranges (Top/Bottom-50%) if no candidates are available. Each valid pair yields a triplet $(q, \tau^+, \tau^-)$ used for DPO training.

For clarity, we provide in the main text a simplified version of the algorithm, while a fully detailed pseudocode with all implementation nuances and fallback rules is presented in Appendix, ensuring reproducibility and transparency of our method.

---

**Algorithm 1:** Simplified Per-Prompt DPO Pair Construction

**Input:** Recent prompts $\mathcal{H}$, candidate sets $\{\mathcal{C}_q\}$
**Output:** Triplets $(q, \tau^+, \tau^-)$
Select recent $L$ prompts from $\mathcal{H}$;
**foreach** *prompt $q$* **do**
     Filter candidates $\mathcal{C}_q^{\mathcal{F}}$;
     Pick $\tau^+$ from high-score pool with similarity in $I_1 \rightarrow I_2 \rightarrow I_3 \rightarrow$ Top-50%;
     Pick $\tau^-$ from low-score pool with similarity in $I_1 \rightarrow I_2 \rightarrow I_3 \rightarrow$ Bottom-50%;
     Record triplet $(q, \tau^+, \tau^-)$;

---

**Dataset and hyperparameters.** Let $L$ be the number of recent prompts used and $r$ the number of pairs per prompt (default $r = 1$). The resulting DPO dataset size is at most $D \leq L \cdot r$. The key hyperparameters are $\alpha$ (score quantile, default $0.3$), the similarity relaxation windows $I_1, I_2, I_3$, and the global similarity acceptance band $\mu \pm \sigma$. These are chosen to (i) favor high-quality examples, (ii) ensure chosen/rejected pairs are structurally comparable, and (iii) avoid pairing identical prompt with widely varying responses that confuse the learner.

**Why this reduces distribution shift.** By (a) computing $\mu, \sigma$ from the union of both models' outputs, (b) enforcing the global similarity filter $\mathcal{F}$, and (c) selecting chosen/rejected candidates from narrow similarity windows, we ensure that the training pairs are consistent with the local model's typical output distribution. This reduces the likelihood that the local model is asked to map a single prompt to mutually contradictory responses and therefore stabilizes DPO optimization.

## 5 EXPERIMENTS

### 5.1 EXPERIMENTAL SETUP

We evaluate MCCE in the domain of multi-objective drug design, a highly challenging problem that requires navigating an enormous chemical space to identify molecules balancing multiple, often conflicting, properties. For the frozen, closed-source LLMs, we leveraged the GPT-4o-2024-05-13 and Gemini-2.5-flash-nothinking models through their APIs, while the local trainable component was instantiated with Qwen2.5-7B-Instruct. The initial population of candidate molecules was constructed by randomly sampling 100 molecules from the ZINC dataset, ensuring sufficient diversity at the start of evolution. The generated molecules were assessed against five standard drug-likeness objectives, and the final optimization outcome was measured using the Hypervolume Indicator (HV), a widely adopted metric in multi-objective optimization that jointly reflects solution quality and diversity. For training paradigms, we implemented SFT and RL baselines using the `verl` library, while our DPO method was implemented with the `trl` library to ensure stable preference-based optimization.

### 5.2 MAIN RESULTS

#### 5.2.1 OVERALL PERFORMANCE

Table 1 provides a comprehensive evaluation across three critical dimensions. First, in terms of Internal Mechanisms, our DPO-driven approach significantly outperforms both single-model baselines and alternative training paradigms (SFT and RL). It achieves the highest Top-1 Fitness by effectively mitigating catastrophic forgetting and training instability. Second, compared against SOTA Baselines such as GFlowNet and DyMol, MCCE demonstrates superior optimization capability. It consistently discovers molecules with higher fitness while maintaining competitive diversity and validity scores. Finally, the Ablation Studies validate the optimality of our design choices, confirming that the proposed similarity-based data synthesis ($\alpha = 0.30$), asymmetric collaboration split (50/32), and frequent model updates are essential factors for maximizing system performance.Figure 2 and Table 1 present a clear comparison of the collaborative system's performance with and without parameter training, unequivocally demonstrating that the continuous learning mechanism is crucial for long-term optimization gains.

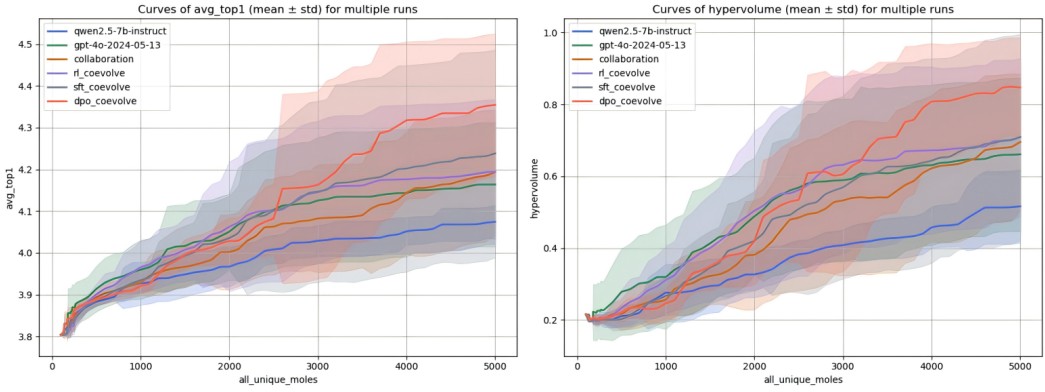

Figure 2: Overall performance comparison across different baselines.(Left) The curve of **avg_top1** (mean ± std) shows that our DPO-enhanced co-evolutionary framework consistently outperforms all baselines, steadily increasing the average quality of the top-ranked molecule throughout the optimization process.(Right) The curve of **hypervolume** (mean ± std) further highlights the superiority of our approach: MCCE with DPO training achieves the largest Pareto front coverage, demonstrating both improved solution quality and diversity.In both metrics, our method significantly surpasses single-model baselines (e.g., Qwen2.5-7B-Instruct, GPT-4o-2024-05-13) as well as alternative co-evolution variants (SFT and RL), achieving state-of-the-art performance.

Table 1: Comprehensive performance comparison on multi-objective optimization tasks. The table is categorized into: (1) **Internal Mechanisms & Training Paradigms**, analyzing the impact of different learning strategies and model components; (2) **Comparison with SOTA Baselines**; and (3-5) Detailed **Ablation Studies** on similarity, split ratio, and update frequency. Results are reported as mean $\pm$ std over 5 runs. Best results within each section are in **bold**, and second-best are underlined.

| Model | Top1 F | Top10 F | AUC-Top10 | HV | Diversity | Uniqueness | Validity |
|---|---|---|---|---|---|---|---|
| *Internal Mechanisms & Training Paradigms* | | | | | | | |
| qwen2.5-7b-instruct | 4.07 ± 0.04 | 4.05 ± 0.04 | 3.91 ± 0.03 | 0.516 ± 0.102 | 0.543 ± 0.047 | 0.576 ± 0.018 | 0.838 ± 0.025 |
| gpt-4o-2024-05-13 | 4.16 ± 0.15 | 4.14 ± 0.12 | 3.99 ± 0.08 | 0.661 ± 0.214 | 0.497 ± 0.035 | 0.702 ± 0.056 | 0.902 ± 0.022 |
| collaboration | 4.19 ± 0.15 | 4.13 ± 0.12 | 3.96 ± 0.06 | 0.695 ± 0.189 | 0.524 ± 0.048 | 0.750 ± 0.041 | 0.838 ± 0.024 |
| rl_coevolve | 4.19 ± 0.17 | 4.16 ± 0.15 | 3.99 ± 0.09 | 0.709 ± 0.219 | 0.509 ± 0.059 | 0.683 ± 0.045 | 0.893 ± 0.021 |
| sft_coevolve | 4.24 ± 0.25 | 4.20 ± 0.22 | 3.99 ± 0.11 | 0.709 ± 0.288 | 0.478 ± 0.070 | 0.571 ± 0.047 | 0.905 ± 0.020 |
| **MCCE (dpo_coevolve)** | **4.35 ± 0.17** | **4.28 ± 0.15** | **4.02 ± 0.09** | 0.847 ± 0.138 | 0.484 ± 0.063 | 0.660 ± 0.018 | 0.820 ± 0.022 |
| - *dpo_coevolve:local* | 4.27 ± 0.16 | 4.22 ± 0.14 | 4.01 ± 0.03 | 0.826 ± 0.126 | **0.555 ± 0.055** | 0.633 ± 0.025 | 0.759 ± 0.030 |
| - *dpo_coevolve:api* | **4.35 ± 0.17** | 4.28 ± 0.14 | 4.03 ± 0.07 | **0.855 ± 0.135** | 0.505 ± 0.062 | **0.784 ± 0.016** | **0.907 ± 0.016** |
| *Comparison with SOTA Baselines* | | | | | | | |
| GB-GA(Jensen, 2019) | 4.02 ± 0.10 | 3.98 ± 0.10 | 3.86 ± 0.05 | 0.643 ± 0.268 | 0.623 ± 0.047 | 0.821 ± 0.032 | **1.000 ± 0.000** |
| REINVENT(Olivecrona et al., 2017) | 4.23 ± 0.20 | 4.14 ± 0.22 | 3.93 ± 0.13 | 0.742 ± 0.259 | 0.640 ± 0.111 | 0.690 ± 0.132 | 0.979 ± 0.002 |
| MoLLEO(Wang et al., 2024) | 4.19 ± 0.08 | 4.08 ± 0.02 | 3.95 ± 0.02 | 0.860 ± 0.088 | **0.670 ± 0.015** | 0.575 ± 0.075 | 0.938 ± 0.007 |
| GFlowNet(Kim et al., 2024) | 4.24 ± 0.25 | 4.20 ± 0.21 | **4.08 ± 0.15** | **0.871 ± 0.088** | 0.633 ± 0.066 | 0.349 ± 0.004 | 0.998 ± 0.000 |
| DyMol(Shin et al., 2024) | 4.23 ± 0.17 | 4.16 ± 0.13 | 4.00 ± 0.05 | 0.868 ± 0.146 | 0.581 ± 0.069 | **0.986 ± 0.005** | **1.000 ± 0.000** |
| **MCCE (Ours)** | **4.35 ± 0.17** | **4.28 ± 0.15** | 4.02 ± 0.09 | 0.847 ± 0.138 | 0.484 ± 0.063 | 0.660 ± 0.018 | 0.820 ± 0.022 |
| *Ablation: Similarity Strategy* | | | | | | | |
| dpo_coevolve ($\alpha = 0.30$) | 4.35 ± 0.17 | **4.28 ± 0.15** | 4.02 ± 0.09 | 0.847 ± 0.138 | 0.484 ± 0.063 | 0.660 ± 0.018 | 0.820 ± 0.022 |
| dpo_coevolve ($\alpha = 0.30$, only_I3) | 4.32 ± 0.14 | 4.21 ± 0.10 | 3.99 ± 0.08 | **0.848 ± 0.086** | 0.532 ± 0.059 | **0.694 ± 0.058** | 0.829 ± 0.022 |
| dpo_coevolve ($\alpha = 0.40$) | **4.36 ± 0.12** | 4.27 ± 0.07 | 3.99 ± 0.05 | 0.843 ± 0.091 | 0.462 ± 0.009 | 0.664 ± 0.072 | 0.816 ± 0.025 |
| dpo_coevolve ($\alpha = 0.20$) | 4.33 ± 0.11 | 4.26 ± 0.06 | 4.02 ± 0.06 | 0.828 ± 0.091 | 0.483 ± 0.030 | 0.651 ± 0.052 | **0.836 ± 0.047** |
| dpo_coevolve (embedding) | 4.29 ± 0.12 | 4.22 ± 0.07 | 3.98 ± 0.05 | 0.847 ± 0.104 | 0.515 ± 0.060 | 0.688 ± 0.038 | 0.825 ± 0.023 |
| *Ablation: Call Split (API/Local)* | | | | | | | |
| 50/50 | 4.31 ± 0.12 | 4.22 ± 0.08 | 3.99 ± 0.05 | 0.838 ± 0.120 | 0.453 ± 0.061 | 0.641 ± 0.082 | 0.816 ± 0.038 |
| 50/32 | **4.35 ± 0.17** | **4.28 ± 0.15** | **4.02 ± 0.09** | **0.847 ± 0.138** | **0.484 ± 0.063** | **0.660 ± 0.018** | **0.820 ± 0.022** |
| 50/16 | 4.25 ± 0.20 | 4.19 ± 0.15 | 3.96 ± 0.06 | 0.734 ± 0.249 | 0.462 ± 0.015 | 0.656 ± 0.111 | 0.809 ± 0.056 |
| *Ablation: Update Frequency* | | | | | | | |
| 500 candidates | 4.28 ± 0.17 | 4.21 ± 0.12 | 3.98 ± 0.06 | 0.796 ± 0.205 | 0.493 ± 0.054 | 0.658 ± 0.099 | **0.848 ± 0.048** |
| 200 candidates | 4.32 ± 0.17 | 4.27 ± 0.16 | **4.02 ± 0.11** | **0.856 ± 0.160** | 0.500 ± 0.055 | 0.626 ± 0.033 | 0.823 ± 0.019 |
| 1_round | **4.35 ± 0.17** | **4.28 ± 0.15** | 4.02 ± 0.09 | 0.847 ± 0.138 | 0.484 ± 0.063 | **0.660 ± 0.018** | 0.820 ± 0.022 |

### 5.2.2 The Co-evolutionary Curve and Output Distribution Analysis

To highlight the effectiveness of our collaborative design, we present two complementary visualizations in Figure 3.

**(Left) The co-evolutionary curve.** This curve captures the dynamics of how the frozen large LLM and the fine-tuned local model collaborate throughout the optimization process. The large LLM consistently provides broad global exploration, generating diverse candidates guided by its rich prior knowledge. In parallel, the local model—refined through iterative learning from breakthrough trajectories—adapts to the search space and performs targeted exploitation. The alternating interplay between these two roles prevents premature convergence, increases diversity, and steadily drives the optimization toward superior regions of the search space. The curve clearly illustrates that their collaboration outperforms the trajectory of either model alone.

**(Right) Output distribution analysis.** To further examine the learning effect, we analyze the quality distribution of molecules generated by three models: the frozen LLM, the initial (untrained) local model, and the fine-tuned local model. Using a no-parent prompt, we sample 1,000 molecules from each model. The histogram shows that the trained local model produces a distribution shifted significantly toward higher scores, surpassing both the frozen LLM and the untrained local baseline. This confirms that the fine-tuning procedure successfully internalizes experience, allowing the local model to approximate the distribution of high-quality molecules. Combined with the steadily decreasing training loss, this analysis demonstrates that our framework not only generates strong solutions but also achieves continual improvement through experience-driven learning.

### 5.3 Generalization to Combinatorial Optimization

To further validate the universality of the MCCE framework beyond molecular design, we extended our evaluation to three classic NP-hard problems: the Circle Packing problem, the Multi-Objective Traveling Salesman Problem (MOTSP), and the Multi-Objective Capacitated Vehicle Routing Problem (MOCVRP).

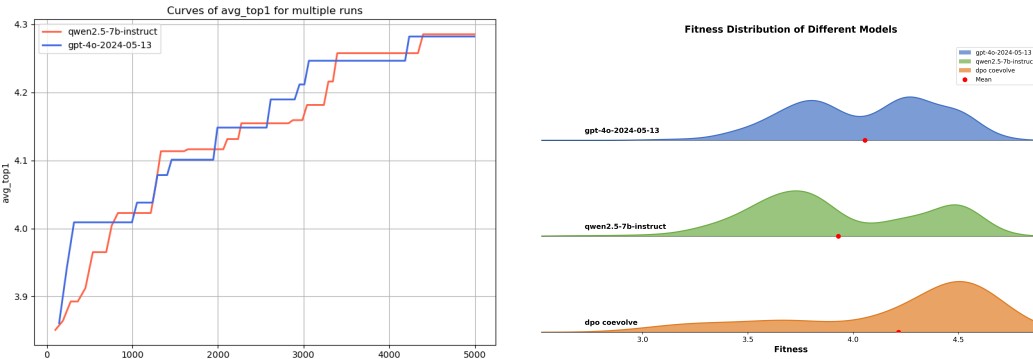

Figure 3: (Left) The co-evolutionary curve showing how the large LLM and local model complement each other to achieve superior trajectories. (Right) Output distribution analysis of molecules generated from the frozen LLM, the initial local model, and the fine-tuned local model.

**Circle Packing:** Requires placing $n$ non-overlapping circles in a unit square to maximize the common radius. We compare our results against long-standing community records.

**MOTSP:** Seeks a single Hamiltonian circuit that, starting and ending at a given depot, visits every city exactly once while simultaneously minimizing multiple conflicting objectives.

**MOCVRP:** Designs a set of capacitated vehicle routes that originate from a common depot, serve all customer demands, and jointly minimize the total travel distance and the makespan.

Benchmark instances for the combinatorial tasks were generated following Lin et al. (2022). We adopt Hyper-volume (HV) as the performance indicator. Baselines include the solver Pymoo (Blank & Deb, 2020) and recent search-based algorithms such as ReEvo, AlphaEvolve, AIDE (Jiang et al., 2025), and FunSearch. The results are shown in Table 2 and Table 3.

Table 2: Hypervolume comparison on Combinatorial Optimization. MCCE achieves SOTA performance on MOCVRP and remains highly competitive on MOTSP against recent baselines.

| Method | MOTSP ($n = 100$) | MOCVRP ($n = 100, m = 20$) |
|---|---|---|
| Pymoo | 0.983488 | 0.955802 |
| AIDE | 1.020798 | 1.005552 |
| FunSearch | 1.023301 | 1.032126 |
| ReEvo | 1.028890 | 1.034541 |
| AlphaEvolve | **1.029279** | 1.031803 |
| **MCCE** | 1.025206 | **1.048843** |

Table 3: Comparison of MCCE against current community records for the Circle Packing problem. MCCE successfully discovers configurations surpassing the previous best-known results.

| Size | Current Record | MCCE |
|---|---|---|
| $n = 26$ | 2.634+ | **2.635983** |
| $n = 31$ | 2.889+ | **2.889970** |

## 6 CONCLUSION

In this work, we presented MCCE, a collaborative co-evolutionary framework that unites a frozen large language model with a trainable local model to tackle large-scale multi-objective discrete optimization. Our approach establishes a closed feedback loop where the LLM drives global exploration while the local model progressively improves through experience-driven learning, yielding a mutually reinforcing synergy rather than one-way distillation. Extensive experiments in multi-objective drug design demonstrate that this hybrid paradigm achieves state-of-the-art performance and significantly surpasses existing baselines. Beyond its empirical success, MCCE highlights a broader principle: hybrid AI systems that combine powerful static models with adaptive, trainable counterparts can unlock new capabilities in complex problem-solving. Looking forward, we envision extending MCCE to other domains of discrete optimization and exploring more adaptive mechanisms for inter-model communication and dynamic balance, further strengthening the generality and impact of this paradigm.

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

## A  APPENDIX

### A.1  PROMPT EXAMPLE

Suggest new molecules that satisfy the following requirements: 1. decrease the SA value. 2. decrease the DRD2 value. 3. increase the QED value. 4. decrease the GSK3ŏ3b2 value. 5. increase the JNK3 value.

sa: SA measures how easily a molecule can be synthesized based on its structural complexity. Simplifying a molecule by reducing complex ring systems or functional groups can lower SA, making synthesis easier, while adding complex structures can increase SA, making synthesis harder.

drd2: Dopamine receptor D2 (DRD2) is a receptor involved in the modulation of neurotransmission and is a target for various psychiatric and neurological disorders. Adding functional groups like hydroxyl or halogen atoms to aromatic rings can enhance binding affinity to DRD2. Removing aromaticity or introducing bulky groups near the binding sites often decreases DRD2 activity.

qed: QED (Quantitative Estimate of Drug-likeness) is a measure that quantifieshow 'drug-like' a molecule is based on properties such as molecular weight,solubility, and the number of hydrogen bond donors and acceptors.Adding functional groups that improve drug-like properties (e.g., small molecular size,balanced hydrophilicity) can increase QED, while introducing large, complex, or highly polar groups can decrease it.

gsk3b: Glycogen synthase kinase-3 beta (GSK3ŏ3b2) is an enzyme involved in cellular processes like metabolism and apoptosis, and is a therapeutic target for cancer and neurological diseases.Adding polar groups, such as hydroxyls, can improve hydrogen bonding with GSK3ŏ3b2's active site.Introducing steric hindrance or highly hydrophobic regions can reduce interactions with GSK3ŏ3b2.

jnk3: c-Jun N-terminal kinase 3 (JNK3) is a kinase involved in stress signaling and is targeted for neuroprotection in diseases like Alzheimer's.Introducing small polar or electronegative groups can enhance binding affinity to JNK3.Removing polar functional groups or adding large, bulky substituents can reduce activity by obstructing the active site.

Give me 2 new molecules that fit the features.

You can do it by applying crossover on the given points and based on your knowledge. The molecule should be valid.

Do not write code. Do not give any explanation. Each output new molecule must start with mol and end with /mol in SIMLES form.Your answer can only contain two molecules and end immediately.

### A.2  DPO LOSS ANALYSIS

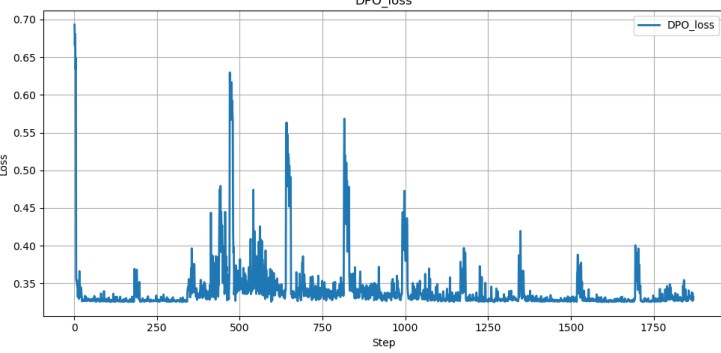

Figure 4: loss Analysis

Figure 4 plots the training loss curve of our DPO optimization. We observe that as the training step increases, the overall loss gradually decreases and the peak values become progressively lower.

This trend indicates that the local model is steadily learning and aligning with the distribution of high-quality molecules. The occasional sharp peaks correspond to the introduction of newly synthesized training data, which temporarily increases the difficulty of optimization. Importantly, the diminishing magnitude of these peaks over time reflects that the model is effectively absorbing new knowledge while maintaining stability, thereby confirming the robustness of our similarity-based data synthesis strategy.

## A.3 TRAINING DETAILS

To ensure stability during Direct Preference Optimization (DPO) training, we adopt the *initial untrained local model* $\pi_{\theta_0}$ as the reference model $\pi_{\mathrm{ref}}$ in the DPO loss. For each training triplet $(q, \tau^+, \tau^-)$, the DPO objective is

$$\ell_{\mathrm{DPO}}(q, \tau^+, \tau^-) \;=\; -\log \sigma \left( \beta \Big[ \log \frac{\pi_\theta(\tau^+ \mid q)}{\pi_{\theta_0}(\tau^+ \mid q)} - \log \frac{\pi_\theta(\tau^- \mid q)}{\pi_{\theta_0}(\tau^- \mid q)} \Big] \right), \tag{15}$$

where $\sigma(\cdot)$ is the sigmoid function and $\beta > 0$ is a scaling parameter. By fixing $\pi_{\mathrm{ref}} = \pi_{\theta_0}$, we prevent drift of the reference distribution and guarantee that the optimization process always measures progress relative to the original model. This prevents instability that might occur if the reference model itself were updated during training. In practice, we observe a monotonically decreasing average loss curve, which provides evidence that the local model is gradually aligning with the distribution of high-quality molecules.

**Training frequency and dataset size.** We denote by $f$ the update frequency (number of generated candidates between two training updates) and by $|\mathcal{D}|$ the size of the synthesized DPO dataset. Both hyperparameters significantly influence stability and performance. Empirically, smaller $f$ (i.e., more frequent updates) accelerates adaptation but may introduce variance due to limited data per update, while larger $|\mathcal{D}|$ provides smoother gradients at the cost of slower responsiveness.

**Comparison across paradigms.** We conducted extensive hyperparameter sweeps for several training paradigms, including SFT, offline RL, GFlowNets, and our DPO method. Let $\mathcal{M}$ denote the set of all hyperparameter configurations explored for a given method $m$. The optimal performance is reported as

$$\mathrm{Perf}(m) = \max_{\lambda \in \mathcal{M}} \; \mathbb{E}[s(c) \mid c \sim \pi_{m,\lambda}], \tag{16}$$

where $s(c)$ is the evaluation score of molecule $c$ and $\pi_{m,\lambda}$ is the trained model with hyperparameter configuration $\lambda$. Across all settings, our DPO-based approach consistently achieved higher $\mathrm{Perf}(m)$ than SFT and offline RL, and demonstrated greater robustness to hyperparameter variations.

## A.4 DETAILED ALGORITHM FOR SIMILARITY-BASED DATA SYNTHESIS

For completeness, we provide the full pseudocode of the per-prompt DPO pair construction procedure, including all fallback rules and implementation details.

---

**Algorithm 2:** Per-Prompt DPO Pair Construction with Fallback Rules

---

**Input:** Historical prompts $\mathcal{H}$, candidate sets $\{\mathcal{C}_q\}$, similarity filter $\mathcal{F}$, high/low score pools
$\qquad \mathcal{T}_{\text{high}}, \mathcal{T}_{\text{low}}$, intervals $I_1, I_2, I_3$, max recent prompts $L$, max pairs per prompt $r$
**Output:** Set of DPO triplets $\{(q, \tau^+, \tau^-)\}$
Select the most recent $L$ prompts from $\mathcal{H}$;
**foreach** *prompt q in selected prompts* **do**
    Initialize $\mathcal{C}_q^{\mathcal{F}} \leftarrow \mathcal{C}_q \cap \mathcal{F}$;
    **for** $i \leftarrow 1$ **to** $r$ **do**
        Try to select $\tau^+$ from $\mathcal{C}_q^{\mathcal{F}} \cap \mathcal{T}_{\text{high}} \cap I_1$;
        **if** $\tau^+$ *not found* **then**
            Relax to $I_2$;
            If still none, relax to $I_3$;
            If still none, broaden to Top-50% pool;
        **else**
            Keep $\tau^+$
        If multiple candidates satisfy, choose highest-scoring or sample uniformly;
        Try to select $\tau^-$ from $\mathcal{C}_q^{\mathcal{F}} \cap \mathcal{T}_{\text{low}} \cap I_1$;
        **if** $\tau^-$ *not found* **then**
            Relax to $I_2$, then $I_3$, then Bottom-50% pool;
        **else**
            Keep $\tau^-$
        **if** $\tau^+$ *or* $\tau^-$ *missing* **then**
            Optionally skip this prompt or draw a random sample from the respective 50% pool;
        Record triplet $(q, \tau^+, \tau^-)$ and optionally store $s(\tau^{\pm})$, $\text{sim}(\tau^{\pm}, q)$;

---

## A.5 ADDITIONAL CO-EVOLUTIONARY CURVES

To further validate the effectiveness of our collaborative co-evolution framework, we provide four additional co-evolutionary curves in Figure 5. These curves consistently demonstrate the same trend observed in the main text: the frozen large LLM provides broad exploration by leveraging its prior knowledge, while the local model—progressively refined through experience learning—contributes focused exploitation and adaptation to the evolving search space. The alternating interplay between exploration and exploitation prevents premature convergence, enhances diversity, and steadily drives the optimization toward superior solutions. Importantly, across all cases, the collaborative trajectory consistently surpasses that of either model operating alone, confirming the robustness and generality of the co-evolutionary mechanism.

## A.6 SUPPLEMENTARY ANALYSIS OF SYNTHETIC ACCESSIBILITY

To assess the practical applicability of the molecules generated by our MCCE framework, we rigorously evaluated their Synthetic Accessibility (SA). We utilized the SA Oracle provided by the *Therapeutics Data Commons* (TDC)[1], instantiated via `Oracle(name = 'SA')`.

The SA score estimates the difficulty of synthesizing a given molecule based on a combination of the molecule's fragment contributions and molecular complexity. The metric is calculated via RDKit using a set of chemical rules originally defined by Ertl & Schuffenhauer (2009). This implementation is widely adopted in molecular generation benchmarks, including the Molecular Sets (MOSES) platform (Polykovskiy et al., 2020).

The scoring scale follows these empirical guidelines:

- **1.0 – 3.0:** Easy to synthesize.
- **3.0 – 6.0:** Intermediate to difficult.
- **> 6.0:** Very difficult or impossible to synthesize.

---

[1]https://tdcommons.ai/functions/oracles/#synthetic-accessibility-sa

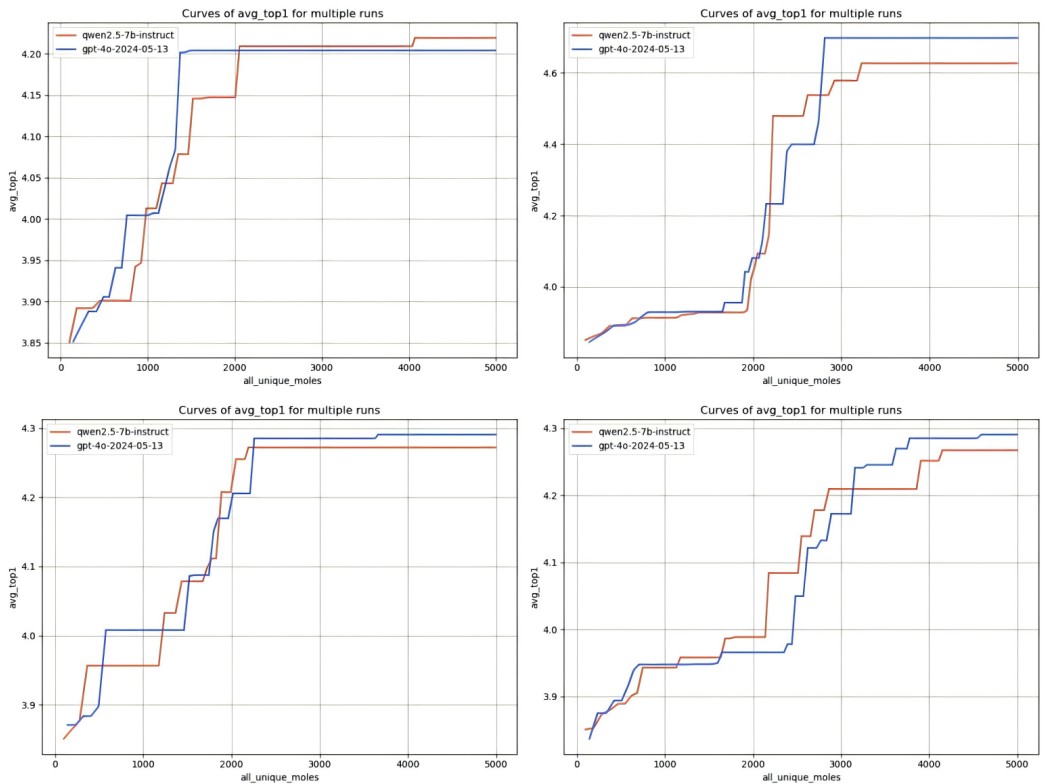

Figure 5: Additional Co-evolutionary Curves

A lower score indicates better synthesizability, which is a critical constraint in real-world drug discovery pipelines.

We conducted a statistical analysis on the final populations ('final_pops') obtained after 5 independent evolutionary runs. The aggregated statistics for the Synthetic Accessibility scores are presented in Table 4.

Table 4: Statistical Analysis of Synthetic Accessibility (SA) Scores in Final Populations.

| Metric | Count ($N$) | Mean | Std. Dev. | Min | Max |
|---|---|---|---|---|---|
| Synthetic Accessibility | 250 | 2.00 | 0.22 | 1.49 | 3.06 |

As shown in Table 4, the generated molecules exhibit exceptional synthesizability profiles:

**High Synthesizability:** The mean SA score is **2.00**, which falls well within the "easy to synthesize" range (1–3). This is significantly lower than the typical threshold for intermediate difficulty, suggesting that the generated candidates are chemically realistic and practical for wet-lab synthesis.

**Concentrated Distribution:** The standard deviation is low (**0.22**), and the range is narrow ([1.49, 3.06]). This indicates that the MCCE framework maintains a tight control over the chemical complexity of the population. Unlike traditional evolutionary methods that might exploit scoring functions by generating overly complex or chaotic structures, our collaborative co-evolution approach effectively filters out "hard-to-synthesize" outliers.

**Successful Multi-Objective Constraint:** It is important to note that these favorable SA scores were achieved while simultaneously optimizing four other challenging objectives (DRD2, QED, GSK3$\beta$, and JNK3). The fact that the maximum SA score observed is only **3.06** (borderline intermediate)

demonstrates that MCCE successfully treats SA as a hard constraint. The local trainable model, refined via DPO, appears to have internalized the implicit rules of chemical validity and simplicity, avoiding the generation of unrealistic high-scoring artifacts.

In conclusion, the SA analysis confirms that the MCCE framework produces high-quality molecular candidates that are not only theoretically potent (high binding affinity) but also practically viable for synthesis.

## A.7 COST ANALYSIS

To demonstrate the cost-effectiveness of the MCCE framework, we tracked the detailed computational resources and financial costs associated with a standard evolutionary run. The experiment was conducted using the **GPT-4o-2024-05-13** model as the frozen global explorer and a local trainable model **Qwen2.5-7B-Instruct** on a high-performance compute node.

The breakdown of the computational budget and incurred costs for generating a total of 5,000 candidates ('Budget_candidates') is summarized in Table 5.

Table 5: Computational Cost and Resource Usage for MCCE (GPT-4o + Local Model). Statistics are reported as Mean $\pm$ Std over multiple runs.

| Metric | Value / Specification |
|---|---|
| **Target Population Budget** | 5,000 Candidates |
| **Model Configuration (API / Local)** | 50 / 32 |
| **Hardware Infrastructure** | $8 \times$ NVIDIA A800 (40GB) |
| **Total LLM Calls** | $3471.14 \pm 165.34$ |
| **Total Wall-clock Time (Hours)** | $3.12 \pm 0.31$ |
| **Total API Cost (USD)** | **\$3.814 $\pm$ 0.457** |

The data in Table 5 highlights several key advantages of the collaborative co-evolution paradigm:

**High Cost-Efficiency:** The total financial cost for accessing the proprietary closed-source model (GPT-4o) was remarkably low, averaging approximately **\$3.81 per run**. This is significantly more economical than pure API-based evolutionary methods, which typically require extensive token consumption for every generation step. By offloading a significant portion of the localized search and exploitation to the local trainable model, MCCE drastically reduces the dependency on expensive API calls.

**Reasonable Time Complexity:** Utilizing an $8\times$A800 GPU cluster, the entire evolutionary process (including DPO fine-tuning and candidate evaluation) concluded in roughly **3.12 hours**. This demonstrates that the framework is computationally feasible for iterative scientific discovery loops, where rapid turnaround is essential.

**Effective Collaboration:** The total number of LLM calls ($\approx 3,471$) relative to the candidate budget (5,000) suggests an efficient generation strategy. The difference implies that the system effectively utilizes crossover, mutation, and the local model's generation capabilities to expand the population, further optimizing the resource-to-performance ratio.

These findings confirm that MCCE offers a scalable and sustainable path for leveraging Large Language Models in optimization, minimizing the "token tax" usually associated with state-of-the-art foundation models.

## A.8 IMPACT OF SIMILARITY METRICS AND DPO STABILITY ANALYSIS

A core finding of our experiments is that the stability of Direct Preference Optimization (DPO) within a co-evolutionary framework is heavily dependent on the structural similarity of the training pairs. Regardless of the specific mathematical definition of the metric, as long as the metric effectively reflects the pairwise similarity between the chosen ($\tau^+$) and rejected ($\tau^-$) trajectories, it significantly stabilizes the training process.

Our experiments demonstrate that DPO training data synthesis without similarity constraints leads to high variance and gradient conflicts, as the model is forced to compare candidates from disjoint distributions. In contrast, enforcing a similarity constraint ensures that the model learns from consistent, local improvements (i.e., "how to make a good molecule slightly better") rather than confusing global jumps.

Figure 6 visualizes this phenomenon. The loss curve without similarity filtering (a) exhibits severe oscillations and instability, whereas the curve with similarity-based data synthesis (b) remains smooth and converges steadily.

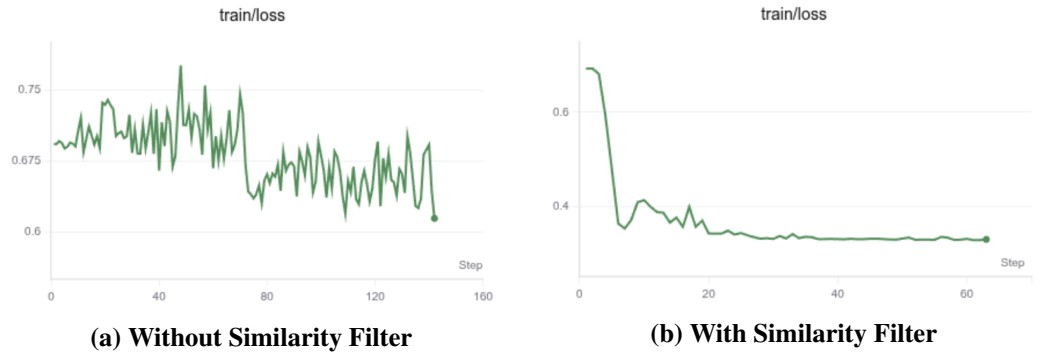

**(a) Without Similarity Filter**      **(b) With Similarity Filter**

Figure 6: Comparison of DPO training stability. **(a)** Without similarity constraints, the loss is highly volatile due to distributional mismatch. **(b)** With our similarity-based data synthesis, the training loss is stable and converges smoothly, confirming the effectiveness of the proposed strategy.

While our primary experiments on molecular optimization utilized domain-specific molecular fingerprints (e.g., Tanimoto similarity on Morgan fingerprints), we extended our framework to other combinatorial and geometric domains by adopting **embedding-based similarity metrics**.

In these supplementary tasks, we found that mapping the decision variables (e.g., routes, coordinates) into a latent embedding space and computing Cosine Similarity yielded equally effective results. This confirms that the MCCE framework is not tied to a specific chemical metric but is a generalizable paradigm dependent only on a reliable measure of "distance" in the solution space. Table 6 details the specific configurations used for each task type.

Table 6: Similarity metrics and configurations used across different optimization tasks. While molecular tasks rely on discrete fingerprints, continuous and combinatorial tasks utilize embedding-based cosine similarity.

| Task Type | Similarity Metric | Embedding Content | Normalization | Similarity Range |
|---|---|---|---|---|
| Circle Packing | Cosine Similarity | Circle centers + radii | L2 normalization | [0, 1] |
| Molecule Optimization | TDC Similarity_Meta | Molecular fingerprints (SMILES) | N/A (external library) | [0, 1] |
| Vehicle Routing | Cosine Similarity | Per-route (customer count, demand, distance) | L2 normalization | [0, 1] |
| Traveling Salesman | Cosine Similarity | Edge length sequences under two objectives | L2 normalization | [0, 1] |

## A.9 DETAILS OF MULTI-OBJECTIVE SELECTION MECHANISM

To strictly balance convergence quality and population diversity—thereby maximizing the Hypervolume (HV) indicator—we designed a **Hybrid Elite-Diversity Selection Strategy**. This strategy constructs the next-generation population of size $N$ by combining Single-Objective (SO) optimization with Pareto-based diversity maintenance.

The population construction is divided into two phases:

- **Elite Preservation (Top 50%):** To ensure rapid convergence towards high-fitness regions, the first half of the population ($N/2$) is selected solely based on the aggregated total score (SO Selection). This acts as a strong exploitation signal.

- **Diversity Maintenance (Bottom 50%):** To prevent the population from collapsing into a single mode and to cover the Pareto front widely, the remaining $N/2$ slots are filled using candidates from the optimal Pareto layers. In this phase, we enforce strict duplicate removal to guarantee structural uniqueness.

The detailed algorithmic flow is as follows:

1. **Elite Selection:** Sort all candidates in the current pool by their aggregated fitness score $S(c)$. Select the top $N/2$ individuals to form the elite set $\mathcal{P}_{elite}$.

2. **Pareto Layering:** Perform Non-Dominated Sorting on the entire candidate pool to partition it into Pareto ranks $\mathcal{R}_1, \mathcal{R}_2, \ldots, \mathcal{R}_k$, where $\mathcal{R}_1$ represents the non-dominated front.

3. **Diversity Filling:** Initialize the diversity set $\mathcal{P}_{div} = \emptyset$. Iterate through ranks $i = 1, 2, \ldots$:
   - Sort candidates within $\mathcal{R}_i$ by total score.
   - Sequentially add candidate $c \in \mathcal{R}_i$ to $\mathcal{P}_{div}$ if and only if $c$ is chemically unique (i.e., its canonical SMILES string is not already present in $\mathcal{P}_{elite} \cup \mathcal{P}_{div}$).
   - Stop once $|\mathcal{P}_{elite}| + |\mathcal{P}_{div}| = N$.

4. **Final Population:** $\mathcal{P}_{next} = \mathcal{P}_{elite} \cup \mathcal{P}_{div}$.

By prioritizing rank-1 Pareto solutions while explicitly filtering duplicates, this method effectively maintains a diverse set of high-quality trade-off solutions, directly contributing to the superior HV and Uniqueness metrics observed in our experiments.

## B  ETHICS STATEMENT

This work adheres to the ICLR Code of Ethics. In this study, no human subjects or animal experimentation was involved. All datasets used, including ZINK, were sourced in compliance with relevant usage guidelines, ensuring no violation of privacy. We have taken care to avoid any biases or discriminatory outcomes in our research process. No personally identifiable information was used, and no experiments were conducted that could raise privacy or security concerns. We are committed to maintaining transparency and integrity throughout the research process.

## C  REPRODUCIBILITY STATEMENT

We have made every effort to ensure that the results presented in this paper are reproducible. All code and datasets have been made publicly available in an anonymous repository to facilitate replication and verification. The experimental setup, including training steps, model configurations, and hardware details, is described in detail in the paper. We have also provided a full description of MCCE, to assist others in reproducing our experiments.

We believe these measures will enable other researchers to reproduce our work and further advance the field.

## D  LLM USAGE

Large Language Models (LLMs) were used to aid in the writing and polishing of the manuscript. Specifically, we used an LLM to assist in refining the language, improving readability, and ensuring clarity in various sections of the paper. The model helped with tasks such as sentence rephrasing, grammar checking, and enhancing the overall flow of the text.

It is important to note that the LLM was not involved in the ideation, research methodology, or experimental design. All research concepts, ideas, and analyses were developed and conducted by the authors. The contributions of the LLM were solely focused on improving the linguistic quality of the paper, with no involvement in the scientific content or data analysis.

The authors take full responsibility for the content of the manuscript, including any text generated or polished by the LLM. We have ensured that the LLM-generated text adheres to ethical guidelines and does not contribute to plagiarism or scientific misconduct.

