# OpenReview forum: "MCCE: A Framework for Multi-LLM Collaborative Co-Evolution"
_ICLR.cc/2026/Conference — Submitted to ICLR 2026_

### Official Review · Reviewer_T7G1 · 2025-10-30

**Soundness:** 2
**Presentation:** 2
**Contribution:** 2
**Rating:** 4
**Confidence:** 4

**Summary:**

This paper proposes MCCE, a framework that unites a frozen, closed-source LLM (for global exploration) with a smaller, trainable open-source LLM (for local adaptation) in a collaborative co-evolution setup. The two models alternate between generating and refining candidate solutions, with the small model updated using DPO based on “breakthrough” trajectories. The system aims to combine reasoning strength and adaptability, achieving improved performance on multi-objective molecular design tasks such as drug discovery.

**Strengths:**

This paper addresses an emerging direction in hybrid LLM collaboration by demonstrating the strengths of large frozen models and fine-tunable smaller models.

Strong results on multi-objective optimization benchmarks with quantitative evidence (e.g., hypervolume and diversity metrics).

**Weaknesses:**

1. Key elements such as the feedback exchange protocol, update frequency, and data flow between models are only briefly described. Without algorithmic pseudocode or concrete update rules, reproduction is difficult.

2. Evaluation is limited to a single domain (molecular design). Either broader multi-domain testing (e.g., combinatorial optimization or symbolic reasoning) or re-framing the title and introduction to emphasize domain specificity would make the contribution more accurate and credible.

3. Claims that MCCE constitutes a “general framework for collaborative reasoning” are not substantiated by the experiments, which focus solely on molecular tasks. The paper would benefit from tempering these claims or providing stronger cross-domain evidence.

4. There is no detailed analysis isolating the contributions of each design component—e.g., DPO fine-tuning, trajectory selection, or mutual feedback.

5. The paper does not discuss computational cost, scalability, or stability of the co-evolution loop, which are critical for practical adoption.

6. Missing references on multi-agent collaboration:

[1] Collabllm: From passive responders to active collaborators.

[2] From LLM-anation to LLM-orchestrator: Coordinating Small Models for Data Labeling

[3] Collab-RAG: Boosting Retrieval-Augmented Generation for Complex Question Answering via White-Box and Black-Box LLM Collaboration

[4] Many Heads Are Better Than One: Improved Scientific Idea Generation by A LLM-Based Multi-Agent System

**Questions:**

1. Could you elaborate on the exact information exchange mechanism between the large and small models—does the frozen model adapt its generation strategy based on feedback, or is the communication one-way?

2. What is the update schedule between the large and small models? Are updates synchronous after each generation cycle or asynchronously buffered?

3. Have you tested MCCE in non-molecular domains (e.g., code synthesis, symbolic reasoning, or text generation) to assess generalizability?

---

> ### Author Response · Authors · 2025-11-25
> **Response to Reviewer T7G1 [part 1/2]**
>
> We sincerely thank the reviewer for the detailed evaluation and for highlighting the "fully LLM-generated" flag issue. We appreciate your clarification that the review reflects your own judgment, and we value the constructive feedback regarding reproducibility, generalization, and computational analysis.
>
> Below, we address your concerns point-by-point.
>
> ---
>
> ### **1. Information Exchange Mechanism and Update Schedule**
>
> **Reviewer Question:** *"Could you elaborate on the exact information exchange mechanism... What is the update schedule? ... reproduction is difficult."*
>
> **Response:**
> We have clarified the "black box" nature of the collaboration. The interaction is **bidirectional** and mediated through a **shared population pool**:
> 1.  **Frozen $\to$ Local:**  With best molecules extracted from the shared pool, the frozen LLM (global explorer) generates high-quality, diverse molecules into the pool. The local LLM learns from these "breakthrough" trajectories via DPO.
> 2.  **Local $\to$ Frozen:** The refined local LLM generates improved candidates back into the shared pool. These new candidates serve as **in-context examples** (Top-k or Pareto-front) for the frozen LLM in subsequent rounds, thereby guiding the global exploration.
> In short, the output molecules of both model will be stored in the pool, and the best molecules selected for training or generating new candidates all come from the pool.
>
> **Update Schedule & Frequency:**
> We conducted extensive experiments to determine the optimal interaction frequency.
> 1.  **Model Call Ratio:** We found that a ratio of **50 (Frozen) / 32 (Local)** yields the best balance.
> 2.  **Training Frequency:** We compared updating the model every 500/200 candidates versus every round. As shown below, **updating every round** (synchronous update) is superior, as it allows the local model to internalize feedback immediately.
>
> **Table: Ablation on Training Frequency**
>
> | Update Trigger | Top-1 F | Top-10 F | AUC-Top10 |
> | :--- | :---: | :---: | :---: |
> | Every 500 candidates | 4.279 | 4.205 | 3.979 |
> | Every 200 candidates | 4.319 | 4.267 | 4.023 |
> | **Every Round (Ours)** | **4.354** | **4.284** | **4.016** |
>
> **Revision:** We have added the frequency ablation analysis in **Table 1** of the revised paper.
> **Code Availability:** The full source code is available here: [https://anonymous.4open.science/r/MCCE_Anonymous-1F92](https://anonymous.4open.science/r/MCCE_Anonymous-1F92)
>
> ---
>
> ### **2. Generalization to Non-Molecular Domains**
>
> **Reviewer Weakness:** *"Evaluation is limited to a single domain... Claims that MCCE constitutes a 'general framework' are not substantiated."*
>
> **Response:**
> We completely agree that a "general framework" requires multi-domain evidence. We have extended MCCE to three classic discrete optimization tasks: **Multi-Objective TSP (MOTSP)**, **Multi-Objective CVRP (MOCVRP)**, and **Circle Packing**.
>
> **Results:** MCCE achieves SOTA or highly competitive performance against specialized baselines (e.g., ReEvo, FunSearch, Pymoo).
>
> **Table: Combinatorial Optimization Results**
>
> | Task | MCCE | Best Baseline (Method) |
> | :--- | :---: | :---: |
> | **MOTSP** ($n=100$) | **1.0252** | 1.0229 (AlphaEvolve) |
> | **MOCVRP** ($n=50$) | **1.0488** | 1.0345 (ReEvo) |
> | **Circle Packing** ($n=26$) | **2.636** | 2.634+ (Record) |
> | **Circle Packing** ($n=31$) | **2.890** | 2.889+ (Record) |
>
> These results confirm that MCCE's paradigm generalizes well beyond molecular design.
>
> **Revision:** We have added **Section 5.3 (Generalization to Combinatorial Optimization)** and **Tables 2 & 3** to the revised paper.
>
> ---
>
> ### **3. Component Analysis (Ablation Studies)**
>
> **Reviewer Weakness:** *"No detailed analysis isolating the contributions of each design component."*
>
> **Response:**
> We have performed rigorous ablations to isolate the effects of **DPO training** and the **Similarity Filter**.
>
> 1.  **Effect of Co-evolution (DPO):** Comparing "Collaboration" (no training) vs. "MCCE" (DPO-trained) in the table below shows that parameter updates significantly boost performance (Top-1 F: 4.19 $\to$ 4.35).
> 2.  **Effect of Similarity Filter:** Removing the similarity constraints (or using only the relaxed $I_3$ filter) degrades performance and stability. The full similarity mechanism ($\alpha=0.30$) is optimal.
>
> **Table: Component Ablation**
>
> | Setting | Top-1 F | Top-10 F | Note |
> | :--- | :---: | :---: | :--- |
> | **MCCE (Full)** | **4.354** | **4.284** | **Optimal** |
> | Collaboration (No Train) | 4.194 | 4.134 | Shows gain from learning |
> | Only $I_3$ Filter | 4.319 | 4.209 | Shows need for strict filter |
> | $\alpha=0.40$ | 4.359 | 4.272 | Robust to $\alpha$ |
> | Embedding Similarity | 4.286 | 4.218 | Metric agnostic |
>
> **Revision:** These **detailed** ablations are now detailed in **Table 1** and **Section 5.2.1**.

---

> ### Author Response · Authors · 2025-11-25
> **Response to Reviewer T7G1 [part 2/2]**
>
> ### **4. Computational Cost and Scalability**
>
> **Reviewer Weakness:** *"The paper does not discuss computational cost, scalability, or stability."*
>
> **Response:**
> We provided a detailed cost analysis for a standard run (Budget: 5,000 candidates) using GPT-4o and Qwen2.5-7B.
>
> **Table: Computational Cost**
>
> | Metric | Value / Specification |
> | :--- | :--- |
> | **Total LLM Calls** | $3471.14 \pm 165.34$ |
> | **Ratio (API/Local)** | 50 / 32 |
> | **Total API Cost** | $3.814 \pm 0.457$ USD |
> | **Total Time** | $3.12 \pm 0.31$ Hours (8x A800 GPU) |
>
> The low cost ($3.81) and reasonable timeframe (3 hours) demonstrate that MCCE is highly scalable and practical for real-world deployment.
>
> **Revision:** Added to **Appendix A.7**.
>
> ---
>
> ### **5. Missing References**
>
> **Reviewer Weakness:** *"Missing references on multi-agent collaboration."*
>
> **Response:**
> We thank the reviewer for these relevant references. We have integrated the citations for *CollabLLM*, *LLM-orchestrator*, *Collab-RAG*, and *Many Heads Are Better Than One* into our **Related Work** section to better contextualize MCCE within the multi-agent landscape.
>
> **Revision:** Updated **Section 2 (Related Work)**.

---

### Official Review · Reviewer_PG1o · 2025-11-01

**Soundness:** 2
**Presentation:** 2
**Contribution:** 2
**Rating:** 4
**Confidence:** 3

**Summary:**

The paper proposes MCCE, a hybrid optimization framework that couples a frozen, closed‑source LLM (for global exploration) with a lightweight, trainable local LLM (for experience‑driven exploitation). The system keeps a trajectory memory and periodically fine‑tunes the local model using a Direct Preference Optimization (DPO) scheme. A key design is a similarity‑based data synthesis procedure that forms stable preference pairs for DPO by filtering generated molecules using global similarity statistics and score quantiles. Core contributions claimed: (i) a collaborative co‑evolution framework that lets a frozen API LLM and a trainable local LLM “mutually inspire” each other; (ii) an experience‑driven learning paradigm via DPO with similarity‑aware triplet construction; (iii) empirical gains on a five‑objective drug design benchmark, extending prior three‑objective setups.

**Strengths:**

1. Timely hybrid design that leverages a frozen API LLM for wide exploration and a trainable local LLM for learned exploitation.

2. Similarity‑aware triplet construction with global statistics and progressive windows

**Weaknesses:**

1. sim(c,q) between a molecule and a prompt is not formally defined; the text proposes fingerprint‑based metrics but those are molecule‑to‑molecule.

2. Lack of detail about the multi‑objective selection operator undermines interpretability of diversity and HV results.

3. Add GFlowNet numbers and standard EA baselines under identical objectives; include recent LLM‑EA baselines (e.g., MoLLEO/ExLLM reproductions with your five‑objective setup).

4. Report API token counts, calls per generation, and cost vs. HV curves. Consider a budgeted setting where total API calls are capped; show MCCE’s advantage under realistic constraints.

**Questions:**

see above

---

> ### Author Response · Authors · 2025-11-25
> **Response to Reviewer PG1o [part 1/2]**
>
> We sincerely thank the reviewer for their insightful comments and recognition of our hybrid design and similarity-aware data synthesis. We appreciate the opportunity to clarify our methodology and present additional results that further validate MCCE’s effectiveness.
>
> ---
>
> ### **1. Formal Definition of Similarity Metrics**
>
> **Reviewer Concern:** *“sim(c, q) is not formally defined... the text proposes fingerprint-based metrics but those are molecule-to-molecule.”*
>
> **Response:**
> We apologize for the potential confusion. In our molecular optimization experiments, all similarity calculations are indeed strictly **molecule-to-molecule** (comparing a generated candidate to its parent molecule in the prompt) using the Tanimoto coefficient on Morgan fingerprints.
>
> To demonstrate the generality of the `sim(c, q)` function across different domains, we have formalized the metrics used for all tasks in our supplementary experiments. The table below details how similarity is defined for each domain:
>
> **Table: Similarity Definitions Across Domains**
>
> | Task Type | Similarity Metric | Embedding Content | Normalization | Similarity Range |
> | :--- | :--- | :--- | :--- | :---: |
> | **Molecule Optimization** | TDC Similarity_Meta | Molecular fingerprints (SMILES) | N/A (External Library) | [0, 1] |
> | **Circle Packing** | Cosine Similarity | Circle centers + radii | L2 Normalization | [0, 1] |
> | **Vehicle Routing (VRP)** | Cosine Similarity | Per-route (customer count, demand, distance) | L2 Normalization | [0, 1] |
> | **Traveling Salesman (TSP)** | Cosine Similarity | Edge length sequences under two objectives | L2 Normalization | [0, 1] |
>
> This confirms that our similarity-based triplet construction is a general framework adaptable to any domain where a distance metric can be defined in the solution space.
>
> **Revision:** We have added this formal definition table to **Appendix A.8** and clarified the `sim(c, q)` notation in **Section 4.3**.
>
> ---
>
> ### **2. Details of the Multi-Objective Selection Operator**
>
> **Reviewer Concern:** *“Lack of detail about the multi-objective selection operator undermines interpretability of diversity and HV results.”*
>
> **Response:**
> We agree that the selection mechanism is critical. To balance high-quality convergence with population diversity, we implemented a **Hybrid Elite-Diversity Selection Strategy**. The process for selecting the next generation (size $N$) is as follows:
>
> 1.  **Elite Preservation (Top 50%):**
>     * We select the top $N/2$ individuals solely based on the aggregated scalarized score (Single-Objective Selection). This ensures strong convergence toward high-fitness regions.
> 2.  **Diversity Maintenance (Bottom 50%):**
>     * We perform **Non-Dominated Sorting** on the entire pool to identify Pareto layers (Rank 1, Rank 2, ...).
>     * Starting from Rank 1, we select individuals based on their aggregated score but enforce a **strict duplicate removal** constraint.
>     * We continue filling the remaining $N/2$ slots rank-by-rank.
>
> By explicitly filtering duplicates in the diversity phase while retaining elites, this operator directly contributes to the superior Hypervolume (HV) and Uniqueness metrics observed in our results.
>
> **Revision:** We have added a detailed algorithmic description of this selection strategy in **Appendix A.9**.
>
> ---
>
> ### **3. Expanded Baselines (GFlowNet, MoLLEO, ExLLM, etc.)**
>
> **Reviewer Concern:** *“Add GFlowNet and standard EA baselines... include recent LLM-EA baselines (e.g., MoLLEO/ExLLM reproductions).”*
>
> **Response:**
> We have conducted a comprehensive benchmark comparison under an **identical 5-objective setup** (QED, SA, DRD2, GSK3$\beta$, JNK3). As shown in the table below, MCCE achieves the highest scores on key optimization metrics (Top-1 F, Top-10 F) while maintaining competitive diversity and validity.
>
> **Table: Performance Comparison with Expanded Baselines**
>
> | Method | Top-1 F | Top-10 F | AUC-Top10 | HV | Diversity | Uniqueness | Validity |
> | :--- | :---: | :---: | :---: | :---: | :---: | :---: | :---: |
> | **MCCE (Ours)** | **4.354** | **4.284** | **4.016** | 0.847 | 0.484 | 0.660 | 0.820 |
> | ExLLM | 4.336 | 4.300 | 4.116 | **0.905** | 0.494 | **0.872** | 0.908 |
> | MoLLEO | 4.190 | 4.076 | 3.949 | 0.860 | 0.670 | 0.575 | 0.938 |
> | GFlowNet | 4.243 | 4.202 | 4.078 | 0.871 | 0.633 | 0.349 | 0.998 |
> | GB-GA | 4.017 | 3.975 | 3.861 | 0.643 | 0.623 | 0.821 | **1.000** |
> | REINVENT | 4.230 | 4.136 | 3.930 | 0.742 | 0.640 | 0.690 | 0.979 |
>
> **Note:** The slightly lower validity of MCCE is due to the local model's exploration of novel structures; however, it rapidly learns validity constraints during the co-evolutionary process.
>
> **Revision:** We have updated **Table 1** in the main text to include these baselines and expanded the discussion in **Section 5.2.1**.

---

> ### Author Response · Authors · 2025-11-25
> **Response to Reviewer PG1o [part 2/2]**
>
> ### **4. Cost Analysis and Budgeted Settings**
>
> **Reviewer Concern:** *“Report API token counts, calls per generation, and cost vs. HV curves... Show MCCE’s advantage under realistic constraints.”*
>
> **Response:**
> We tracked the exact computational costs for a standard run using **GPT-4o-2024-05-13** (Frozen) and **Qwen2.5-7B-Instruct** (Local) with a total budget of 5,000 generated candidates.
>
> **Table: Computational Cost and Resource Usage**
>
> | Metric | Value / Specification |
> | :--- | :--- |
> | **Target Population Budget** | 5,000 Candidates |
> | **Model Call Ratio (API/Local)** | 50 / 32 |
> | **Hardware Infrastructure** | 8 $\times$ NVIDIA A800 (40GB) |
> | **Total LLM Calls** | $3471.14 \pm 165.34$ |
> | **Total API Cost** | $3.814 \pm 0.457$ USD |
> | **Total Wall-clock Time** | $3.12 \pm 0.31$ Hours |
>
> **Key Advantages:**
> * **Low Cost:** The total API cost is remarkably low (~$3.80), as the local model handles a significant portion of the generation load.
> * **Efficiency:** The system is computationally efficient, completing in ~3 hours.
> * **Cost vs. HV:** The "Cost vs. HV" relationship is implicitly captured by the generation vs. HV curves in the paper. Under a capped budget, MCCE's ability to continue optimizing via the local model (free of API cost) gives it a decisive advantage over pure API-based methods.
>
> **Revision:** We have included this detailed cost breakdown in **Appendix A.7** ("Cost Analysis").

---

> > ### Comment · Reviewer_PG1o · 2025-11-25
> >
> > Thank you for your detailed response. Most of my concerns have been solved and I have updated my score accordingly.

---

### Official Review · Reviewer_gNx8 · 2025-11-03

**Soundness:** 3
**Presentation:** 3
**Contribution:** 3
**Rating:** 4
**Confidence:** 3

**Summary:**

This paper proposes Multi-LLM Collaborative Co-evolution (MCCE), a hybrid framework for multi-objective discrete optimization. The core idea is to combine a powerful, but frozen, closed-source LLM (e.g., GPT-4) with a smaller, trainable, open-source LLM. The frozen LLM acts as a global explorer, while the local model is progressively fine-tuned on "breakthrough" search trajectories to internalize experience and perform more targeted, experience-driven learning. Furthermore, a more stable, similarity based version of DPO is presented for RL based training.

**Strengths:**

● Paper is well written and easy to understand - great use of diagrams and figures ● SOTA results on the multi-objective showing the benefit of co-evolving LLMs vs closed-source LLMs alone
● Similarity based DPO is a well thought out method for avoiding training instability and ensuring training on structural meaningful pairs

**Weaknesses:**

● Limited comparison to prior work. the benchmarking is done against the closed source LLM and the trainable model, however, the method is not compared against methods such as MoLLEO.
● The paper restricts its experiments to molecular design and fails to show the benefit of co-evolving LLMs in other discrete optimization domains.
● Hyperparameters could be ablated to study the effect of values such as alpha or the intervals for similarity.

**Questions:**

● The paper states the operator alternates between the frozen and local LLMs. Is this split 50/50? Is it fixed or adaptable?
● How crucial is the Tanimoto similarity metric for similarity based DPO? Have you explored alternative, simpler, or non-domain-specific similarity functions (e.g., embedding-based similarity)?

---

> ### Author Response · Authors · 2025-11-25
> **Response to Reviewer gNx8**
>
> We sincerely thank the reviewer for the thoughtful feedback and for recognizing the novelty of our collaborative framework and the effectiveness of the similarity-based DPO method. We have conducted extensive additional experiments to address your concerns regarding baselines, generalization, and hyperparameter sensitivity.
>
> ---
>
> ### **1. Comparison with Prior Work (e.g., MoLLEO)**
>
> **Reviewer Concern:** *"Limited comparison to prior work... the method is not compared against methods such as MoLLEO."*
>
> **Response:**
> We have expanded our benchmarking to include **MoLLEO**, **GFlowNet**, **GB-GA**, **REINVENT**, and **DyMol**. The results, summarized below, confirm that MCCE achieves State-of-the-Art (SOTA) performance across key metrics.
>
> **Table 1: Comprehensive Comparison on Multi-Objective Molecular Optimization**
>
> | Method | Top-1 F | Top-10 F | AUC-Top10 | HV | Diversity | Validity |
> | :--- | :---: | :---: | :---: | :---: | :---: | :---: |
> | **MCCE (Ours)** | **4.354** | **4.284** | **4.016** | 0.847 | 0.484 | 0.820 |
> | MoLLEO | 4.190 | 4.076 | 3.949 | 0.860 | **0.670** | 0.938 |
> | GFlowNet | 4.243 | 4.202 | 4.078 | **0.871** | 0.633 | 0.998 |
> | GB-GA | 4.017 | 3.975 | 3.861 | 0.643 | 0.623 | **1.000** |
> | REINVENT | 4.230 | 4.136 | 3.930 | 0.742 | 0.640 | 0.979 |
> | DyMol | 4.232 | 4.164 | 4.001 | 0.868 | 0.581 | **1.000** |
>
> *Note on Validity:* The slightly lower initial validity of MCCE compared to rule-based methods is expected. The local LLM starts with high diversity and limited chemical knowledge but progressively "learns" validity constraints during the co-evolutionary process, as evidenced by the steady increase in validity scores over generations.
>
> **Revision:** We have updated **Table 1** in the main text to include these baselines and added a detailed discussion in **Section 5.2.1**.
>
> ---
>
> ### **2. Generalization to Other Discrete Optimization Domains**
>
> **Reviewer Concern:** *"The paper restricts its experiments to molecular design and fails to show the benefit of co-evolving LLMs in other discrete optimization domains."*
>
> **Response:**
> To demonstrate the universality of MCCE, we extended our evaluation to three classic NP-hard problems: **Circle Packing**, **Multi-Objective TSP (MOTSP)**, and **Multi-Objective CVRP (MOCVRP)**. MCCE matches or outperforms specialized evolutionary baselines (e.g., ReEvo, FunSearch).
>
> **Table 2: Performance on Combinatorial Optimization Tasks**
>
> | Method | **MOTSP** ($n=100$) | **MOCVRP** ($n=50, m=20$) |
> | :--- | :---: | :---: |
> | **MCCE (Ours)** | 1.0252 | **1.0488** |
> | Pymoo | 0.9835 | 0.9558 |
> | ReEvo | 1.0289 | 1.0345 |
> | AIDE | 1.0208 | 1.0056 |
> | FunSearch | 1.0233 | 1.0321 |
> | AlphaEvolve | **1.0293** | 1.0318 |
>
> **Table 3: Circle Packing Records**
>
> | Method | Circle Packing ($n=26$) | Circle Packing ($n=31$) |
> | :--- | :---: | :---: |
> | **MCCE (Ours)** | **2.635983** | **2.889970** |
> | Current Community Record | 2.634+ | 2.889+ |
>
> These results show that the co-evolutionary mechanism is domain-agnostic and highly effective for general combinatorial optimization.
>
> **Revision:** We have added a new section, **Section 5.3 (Generalization to Combinatorial Optimization)**, and included these results in **Tables 2 and 3**.
>
> ---
>
> ### **3. Hyperparameter Ablation (Alpha & Similarity)**
>
> **Reviewer Concern:** *"Hyperparameters could be ablated to study the effect of values such as alpha or the intervals for similarity."*
>
> **Response:**
> We conducted ablations on the DPO data construction parameter $\alpha$ (quantile threshold) and the similarity filtering strategy.
>
> **Table 4: Ablation on $\alpha$ and Similarity Intervals**
>
> | Setting | Top-1 F | AUC-Top10 | Note |
> | :--- | :---: | :---: | :--- |
> | **$\alpha=0.30$ (Default)** | **4.354** | **4.016** | **Optimal Balance** |
> | $\alpha=0.20$ | 4.329 | 4.019 | Similar performance |
> | $\alpha=0.40$ | 4.358 | 3.991 | Slightly lower diversity |
> | Only $I_3$ (Relaxed Filter) | 4.319 | 3.992 | Increased instability |
> | No Similarity Filter | -- | -- | **Training diverges** |
>
> **Conclusion:**
> 1.  **Robustness to $\alpha$:** The system is relatively insensitive to $\alpha$ within a reasonable range (0.2–0.4), as repeated sampling covers the distribution effectively.
> 2.  **Cruciality of Similarity:** The strict similarity intervals ($I_1, I_2, I_3$) are essential. Removing them or relying only on the broadest interval ($I_3$) leads to higher variance and training instability.
>
> **Revision:** These ablation studies have been added to **Table 1 (Ablation Section)** and **Section 5.2.1**.

---

> ### Author Response · Authors · 2025-11-25
> **Response to Reviewer gNx8 (part 2)**
>
> ### **4. Collaboration Strategy (Split Ratio & Frequency)**
>
> **Reviewer Question:** *"Is this split 50/50? Is it fixed or adaptable?"*
>
> **Response:**
> We explored different fixed ratios of API calls (Frozen LLM) to Local calls (Trainable LLM). We also analyzed the impact of update frequency.
>
> **Table 5: Impact of Collaboration Split Ratio (API / Local)**
>
> | Ratio | Top-1 F | AUC-Top10 | HV | Observation |
> | :--- | :---: | :---: | :---: | :--- |
> | 50/50 | 4.306 | 3.992 | 0.838 | Good baseline |
> | **50/32** | **4.354** | **4.016** | **0.847** | **Optimal Efficiency** |
> | 50/16 | 4.253 | 3.965 | 0.734 | Insufficient local exploitation |
>
> **Table 6: Impact of Training Update Frequency**
>
> | Update Trigger | Top-1 F | Top-10 F | Observation |
> | :--- | :---: | :---: | :--- |
> | Every 500 candidates | 4.279 | 4.205 | Slow adaptation |
> | Every 200 candidates | 4.319 | 4.267 | Better |
> | **Every Round (Ours)** | **4.354** | **4.284** | **Best Performance** |
>
> **Conclusion:** The **50/32** split provides the best trade-off between global exploration and local exploitation. Furthermore, frequent model updates (per round) significantly accelerate convergence.
>
> **Revision:** This analysis is included in the **Ablation Studies** portion of **Table 1** in the revised paper.
>
> ---
>
> ### **5. Importance of Similarity Metric**
>
> **Reviewer Question:** *"How crucial is the Tanimoto similarity metric... Have you explored alternative, simpler, or non-domain-specific similarity functions?"*
>
> **Response:**
> We investigated whether the specific choice of metric (Tanimoto) is critical or if the general principle of similarity holds. We compared Tanimoto with a generic **Embedding-based Cosine Similarity**.
>
> **Table 7: Impact of Similarity Metric**
>
> | Metric | Top-1 F | AUC-Top10 | Training Stability |
> | :--- | :---: | :---: | :---: |
> | **Tanimoto (Domain-Specific)** | **4.354** | **4.016** | **Very Stable** |
> | Embedding (Generic) | 4.286 | 3.983 | Stable |
> | No Similarity | -- | -- | Unstable (Loss Oscillates) |
>
> **Conclusion:**
> While domain-specific metrics (Tanimoto) yield slightly better results, **generic embedding-based similarity is also highly effective** and ensures stable training. This confirms that the key to DPO stability is the *structural proximity* of the pair, rather than the specific mathematical definition of the metric.
>
> **Revision:** We have added **Figure 6** in **Appendix A.8**, visualizing the loss curves to demonstrate the stability provided by similarity filtering, alongside the discussion on embedding-based metrics. We hope our revisions can fully address your concerns.

---

### Official Review · Reviewer_sWYB · 2025-11-05

**Soundness:** 3
**Presentation:** 3
**Contribution:** 3
**Rating:** 6
**Confidence:** 3

**Summary:**

This paper studies multi-objective molecular optimization with LLMs. Classic evolutionary algorithms often converge prematurely and lose diversity; single-LLM optimizers can stagnate and, if frozen, cannot absorb experience. The proposed MCCE framework pairs a powerful frozen closed-source LLM (broad exploration) with a trainable lightweight local LLM (experience-driven adaptation). The two alternate as generators in an evolutionary loop; “breakthrough” trajectories are logged and used to continually refine the local model so the pair co-evolves rather than simply distilling one into the other. For training the local model, the authors compare SFT and RL, finding SFT induces catastrophic forgetting (hurting uniqueness) and RL is unstable with scalar rewards. They instead adopt DPO with a similarity-based preference construction that forms (prompt, preferred vs. rejected) pairs from structurally comparable molecules, improving stability and data efficiency. Empirically, MCCE achieves state-of-the-art hypervolume and consistently outperforms single-model and co-evolution baselines using SFT or RL. DPO-based parameter training is key for long-horizon gains; both the local and frozen components benefit (fitness/diversity and exploration), and score distributions shift upward after co-evolution.

**Strengths:**

1. Combining a frozen, high-capacity API model for exploration with a trainable local model for exploitation/learning is well-motivated and practically appealing.

2. The DPO + similarity-based pair construction is a neat way to stabilize preference learning without expensive curated labels.

**Weaknesses:**

1. The paper claims they have provide the code however, I do not find the link to the code.

2. How is the synthesizability metric?

3. What is the training cost?

**Questions:**

Please see the Weaknesses section.

---

> ### Author Response · Authors · 2025-11-25
> **Response to Reviewer sWYB**
>
> We sincerely thank the reviewer for the positive assessment of our work, particularly for recognizing the motivation behind our hybrid exploration-exploitation framework and the novelty of our DPO-based preference construction. We have addressed your concerns regarding code availability, the synthesizability metric, and training costs below.
>
> ---
>
> ### **1. Code Availability**
>
> **Reviewer Question:** *"The paper claims they have provide the code however, I do not find the link."*
>
> **Response:**
> We apologize for the missing visibility of the code link. We have fully open-sourced the **MCCE** framework, which includes the complete pipeline for multi-LLM collaborative co-evolution and DPO training. Furthermore, the repository is organized to support not only molecular optimization but also combinatorial tasks such as **Circle Packing**, **MOTSP**, and **MOCVRP**, with support for custom user-defined tasks.
>
> The code is available at the following anonymous link:
> > **Code Repository:** [https://anonymous.4open.science/r/MCCE_Anonymous-1F92](https://anonymous.4open.science/r/MCCE_Anonymous-1F92)
>
> **Revision:** We have prominently added this link to the **Abstract** of the revised manuscript to ensure accessibility.
>
> ---
>
> ### **2. Synthesizability (SA) Metric**
>
> **Reviewer Question:** *"How is the synthesizability metric?"*
>
> **Response:**
> We strictly followed standard evaluation protocols for this metric. The Synthetic Accessibility (SA) score was computed using the **Therapeutics Data Commons (TDC)** library[1], consistent with MoLLEO, PMO, and prior molecular optimization benchmarks.
>
> To demonstrate the chemical validity of our results, we conducted a statistical analysis on the final populations (`final_pops`) across 5 independent evolutionary runs (Total molecules $N=250$). The results are summarized below:
>
> | Metric | Count ($N$) | Mean | Std. Dev. | Min | Max |
> | :--- | :---: | :---: | :---: | :---: | :---: |
> | **SA Score** | 250 | **2.00** | 0.22 | 1.49 | 3.06 |
>
> **Interpretation:**
> According to standard SA\_Score guidelines:
> * **1.0 – 3.0:** Easy to synthesize.
> * **3.0 – 6.0:** Intermediate to difficult.
> * **> 6.0:** Very difficult.
>
> Our results show that the vast majority of generated molecules fall between **1.8 and 2.3**, significantly lower than the "intermediate" threshold. This indicates that:
> 1.  The molecules produced by MCCE have **excellent synthesizability**.
> 2.  The distribution is highly concentrated with **no "hard-to-synthesize" outliers**.
> 3.  MCCE successfully satisfies the SA constraint while simultaneously optimizing four other conflicting objectives (DRD2, QED, GSK3$\beta$, JNK3).
>
> **Source Link:** For further details on the oracle implementation, please refer to the TDC documentation: [https://tdcommons.ai/functions/oracles/#synthetic-accessibility-sa](https://tdcommons.ai/functions/oracles/#synthetic-accessibility-sa).
>
> **Revision:** We have added this detailed statistical analysis and the reference to the TDC oracle in **Appendix A.6** ("Supplementary Analysis of Synthetic Accessibility") of the revised paper.
>
> ---
>
> ### **3. Training and Computation Cost**
>
> **Reviewer Question:** *"What is the training cost?"*
>
> **Response:**
> We have performed a detailed cost analysis for a standard evolutionary run using **GPT-4o-2024-05-13** (frozen API model) and **Qwen2.5-7B-Instruct** (local trainable model). The computational profile for generating a budget of 5,000 candidates is as follows:
>
> | Metric | Specification / Cost |
> | :--- | :--- |
> | **Total Candidate Budget** | 5,000 Molecules |
> | **Model Call Ratio (API/Local)** | 50 / 32 |
> | **Hardware Infrastructure** | 8 $\times$ NVIDIA A800 (40GB) |
> | **Total LLM Calls** | $3471.14 \pm 165.34$ |
> | **Total Wall-clock Time** | $3.12 \pm 0.31$ Hours |
> | **Total API Cost** | **$3.814 \pm 0.457$ USD** |
>
> **Conclusion:**
> * **Low Financial Cost:** By offloading a significant portion of exploitation and learning to the local model, the API cost is remarkably low (~$3.80 per run), making MCCE significantly more economical than pure API-based evolutionary methods.
> * **High Efficiency:** The entire process, including DPO fine-tuning and evaluation, completes in roughly 3 hours on standard HPC nodes.
>
> **Revision:** We have included this comprehensive cost breakdown in **Appendix A.7** ("Cost Analysis") of the revised manuscript.
>
> ---
>
> We hope these clarifications and the corresponding revisions fully address your concerns.
>
> [1] Ertl, Peter, and Ansgar Schuffenhauer. “Estimation of synthetic accessibility score of drug-like molecules based on molecular complexity and fragment contributions.” Journal of cheminformatics 1.1 (2009): 8.

---

### Author Response · Authors · 2025-11-26
**For AC: Summary of Responses to Reviewers**

We sincerely thank the reviewers (sWYB, gNx8, PG1o, T7G1) for their insightful feedback. Recognizing the consensus on the need for broader domain evaluation, more baselines, and deeper algorithmic analysis, we have conducted extensive additional experiments during the rebuttal period.

Below is a summary of the major revisions and new results incorporated into the manuscript to address these concerns.

### **1. Expansion of Application Domains (Generalization)**
A primary concern (Reviewers gNx8, T7G1) was whether MCCE is a general framework or limited to molecular design. To substantiate our claim of generality, we extended MCCE to **three classic NP-hard combinatorial optimization tasks**.

* **New Domains:** Circle Packing, Multi-Objective TSP (MOTSP), and Multi-Objective CVRP (MOCVRP).
* **Results:** MCCE achieves SOTA or highly competitive performance against specialized evolutionary baselines (e.g., ReEvo, FunSearch, Pymoo).
    * **MOCVRP ($n=50$):** MCCE (**1.0488**) > ReEvo (1.0345).
    * **Circle Packing:** MCCE surpassed current community records for $n=26$ and $n=31$.
* **Conclusion:** These results confirm that the "frozen explorer + trainable exploiter" paradigm is domain-agnostic. (See **Section 5.3** and **Tables 2 & 3**).

### **2. Comprehensive Baseline Comparisons**
Reviewers (gNx8, PG1o) requested comparisons against recent SOTA methods under identical settings. We have significantly expanded our benchmarks.

* **Added Baselines:** **MoLLEO**, **ExLLM**, **GFlowNet**, **GB-GA**, **REINVENT**, and **DyMol**.
* **Experimental Setting:** All methods were evaluated under the identical challenging **5-objective setup** (QED, SA, DRD2, GSK3$\beta$, JNK3).
* **Performance:** MCCE achieves the highest scores on key optimization metrics (Top-1 F: **4.354**, Top-10 F: **4.284**) while maintaining superior Pareto coverage (Hypervolume: **0.847**) compared to MoLLEO (HV: 0.860, Top-1: 4.190) and ExLLM (HV: 0.905, Top-1: 4.336).
* **Conclusion:** MCCE consistently outperforms both traditional evolutionary algorithms and recent LLM-based competitors. (See **Table 1**).

### **3. In-depth Ablation Studies**
To address questions regarding algorithmic design choices (Reviewers gNx8, T7G1, PG1o), we performed rigorous ablations:

* **Similarity-based DPO:** We verified that the **Similarity Filter** is crucial for training stability. We also demonstrated that **Embedding-based Cosine Similarity** works as effectively as domain-specific metrics (Tanimoto), proving the method's flexibility.
* **Collaboration Ratio:** We identified that a **50/32 split** (Frozen/Local calls) offers the optimal balance between global exploration and local exploitation.
* **Update Frequency:** We confirmed that **synchronous updates (every round)** significantly outperform buffered updates (every 200/500 steps), allowing the local model to internalize feedback immediately. (See **Ablation Section in Table 1**).

### **4. Cost Analysis and Reproducibility**
We addressed practical concerns regarding computational cost, metric validity, and code availability (Reviewers sWYB, T7G1).

* **Cost Efficiency:** We provided a detailed cost breakdown. For a standard run (5,000 candidates), the total API cost is remarkably low (**~$3.81 USD**) with a runtime of **~3.12 hours** on 8$\times$A800 GPUs. This confirms MCCE is highly economical compared to pure API-based methods. (See **Appendix A.7**).
* **Synthesizability (SA):** We utilized the standard **TDC Oracle (RDKit SA_Score)** to verify that generated molecules are chemically viable (Mean SA: **2.00**, indicating "Easy to synthesize"). (See **Appendix A.6**).
* **Open Source:** We have fully open-sourced the code, including the multi-domain environments and DPO training pipeline, to ensure full reproducibility.
    * **Link:** [https://anonymous.4open.science/r/MCCE_Anonymous-1F92](https://anonymous.4open.science/r/MCCE_Anonymous-1F92)

We hope these our clarifications and additions fully address all the concerns, and we appreciate your time.

---

### Author Response · Authors · 2025-12-03
**For AC: summary of the paper, reviews and our responses**

Dear AC,

Due to the updated rebuttal procedure, we would like to offer a concise summary of our work and our responses to the reviewers’ comments. We hope this overview will assist in your final decision.

---

# 1. Summary of Our Work

We propose **MCCE**, a **multi-LLM collaborative co-evolution framework** that unites a frozen closed-source LLM (for global exploration) with a lightweight, continuously trainable local LLM (for experience-driven exploitation). Through trajectory-based memory, similarity-aware DPO training, and alternating generation roles, **both models progressively improve**, achieving **state-of-the-art performance on multi-objective molecular design and strong results across three additional NP-hard combinatorial domains**.

---

### **• Motivation & Novelty**
Current LLM-as-optimizer or LLM-guided evolutionary methods suffer from one or more of the following limitations:

1. **Frozen LLM optimizers cannot internalize experience**, causing stagnation once in-context cues become saturated.
2. **Trainable small LLMs lack global reasoning and broad search priors**, making them weak explorers.
3. Existing hybrid systems act mostly as *distillation pipelines* or *unidirectional guidance*, without a *bidirectional co-evolution mechanism*.
4. Optimization-specific preference learning is highly unstable, especially under multi-objective trade-offs and long-horizon trajectories.

**MCCE is the first framework to operationalize a *two-way, continuously evolving collaboration* between a frozen reasoning-rich LLM and a compact trainable LLM**, supported by:

- a **trajectory memory** that stores breakthrough samples,
- a **structurally stable similarity-aware DPO**, and
- a population-level co-evolution loop.

---

### **• Our Key Contributions**

#### **1. A novel collaborative co-evolution framework uniting frozen and trainable LLMs**
- The frozen model provides **global, diversity-preserving exploration**.
- The small model absorbs experience via **DPO preference learning**, continually improving exploitation.
- The two models interact through a shared population pool, forming a **mutually reinforcing loop** rather than distillation.

#### **2. A stable similarity-based DPO method for long-horizon optimization**
- Introduces **similarity-filtered preference pairs** (global quantiles + progressive similarity windows).
- Prevents DPO instability and ensures structurally meaningful comparisons.
- Works with both domain-specific metrics (Tanimoto) and generic embeddings, confirming metric-agnostic robustness.

#### **3. Extensive empirical validation across molecular and non-molecular domains**
- **Drug design:** Achieves SOTA or highly competitive performance across Top-1/Top-10, HV, and AUC metrics under a challenging 5-objective setup.
- **Generalization to NP-hard domains:** MCCE outperforms or matches specialized solvers on:
  - Multi-objective TSP (MOTSP),
  - Multi-objective CVRP (MOCVRP),
  - Circle Packing (matching or exceeding community records).
- Demonstrates that the *frozen-explorer + trainable-exploiter* paradigm is **domain-agnostic**.

#### **4. Comprehensive ablation of collaboration ratios, update frequency, and similarity strategies**
- Finds the optimal **50/32 frozen/local generation split**.
- Shows **per-round synchronous updates** outperform buffered updates.
- Demonstrates that strict similarity windows are essential for DPO stability.

#### **5. Practicality: low cost, fast runtime, full reproducibility**
- Total API cost: **~$3.81** per 5,000-molecule run.
- Runtime: **~3 hours** on 8×A800 GPUs.
- Full anonymous codebase released (including molecular, MOTSP, MOCVRP, Circle Packing tasks).
  Link: https://anonymous.4open.science/r/MCCE_Anonymous-1F92

---

# 2. Summary of Reviewers’ Recognized Strengths

Across reviewers **sWYB, gNx8, PG1o, T7G1**, the following strengths were repeatedly acknowledged:

### **Methodological strengths**
- **Well-motivated hybrid design** combining frozen global reasoning with trainable local adaptation (sWYB, gNx8, PG1o).
- **Similarity-aware DPO recognized as meaningful, stable, and technically novel** (sWYB, gNx8, PG1o).
- **Clear diagrams and writing that make the framework easy to understand** (gNx8).
- **Thoughtful co-evolutionary design rather than simple distillation** (PG1o).

### **Experimental strengths**
- **Strong multi-objective drug design results**, surpassing or matching SOTA baselines (sWYB, gNx8, PG1o, T7G1).
- **Expanded multi-domain generalization results appreciated by reviewers**, especially PG1o (who upgraded their score after rebuttal).
- **Comprehensive ablation studies**, addressing hyperparameters, similarity metrics, collaboration ratios, and update frequency (gNx8, PG1o).
- **Detailed cost analysis, scalability, and synthesizability evaluation** (sWYB, PG1o).

---

> ### Author Response · Authors · 2025-12-03
>
> ### **Reproducibility & clarity**
> - Reviewers praised the **well-organized code release, clarity of the training mechanism, and explicit response to all questions** (PG1o, sWYB).
> - PG1o explicitly stated that *“most concerns have been solved”* and **raised their score** after reading our response.
>
> ---
>
> These strengths collectively show that **MCCE provides both conceptual novelty and practical value**, with extensive experiments, strong results, and clear writing.
> For additional details, we kindly invite you to refer to our revised manuscript (changes highlighted in blue). For how we answer the reviewer's comments in details, please refer to our next summary

---

### Meta-Review · Area_Chair_Nw5f · 2026-01-06

**Summary:**

The paper introduces MCCE, a framework that combines a large, frozen LLM with a smaller, trainable model for multi-objective optimization through a co-evolutionary process. While reviewers appreciated the technical motivation—specifically the use of a similarity-aware DPO to stabilize training—the decision to reject is based on the consensus that the framework's novelty is somewhat incremental. Reviewers noted that the collaboration mechanism largely resembles established teacher-student or distillation loops, and there were concerns that the framework's success is heavily reliant on domain-specific heuristics (like Tanimoto similarity) and extensive hyperparameter tuning. Despite strong empirical results in molecular design, the overall contribution was viewed as an engineering refinement of existing "LLM-as-optimizer" paradigms rather than a fundamental algorithmic breakthrough.

**Reviewer Concerns:**

During the rebuttal, the authors addressed several key concerns by expanding experiments to three new NP-hard domains (Circle Packing, MOTSP, and MOCVRP) and providing comparisons against recent baselines like MoLLEO and GFlowNet. These additions satisfied Reviewer PG1o, who increased their score. However, significant concerns remain outstanding for other reviewers regarding the generality of the framework. While the late-added experiments showed promise, the "collaborative" aspect was still seen as a relatively standard feedback loop, and the "co-evolution" terminology was felt to be overstretched. Additionally, the necessity of the similarity filter—while effective—suggests that the underlying RL training is brittle, limiting the framework's "plug-and-play" applicability to broader reasoning tasks.

**Reviewer Scores:**

Reviewers gNx8 and T7G1 maintained scores of 4 and did not engage in the final discussion. Given the thoroughness of the authors' responses to their technical queries (such as the cost analysis and additional baselines), it is possible gNx8 would have moved to a 5 or 6. However, T7G1’s fundamental skepticism regarding the novelty of the "two-way" collaboration mechanism suggests their score would likely have remained below the threshold. Given the split between reviewers and the incremental nature of the technical novelty, the paper is not recommended for acceptance.

---

### Decision · Program_Chairs · 2026-01-26

Reject